# Set Smoothness Unlocks Clarke Hyper-stationarity in Bilevel Optimization

**He Chen**
SEEM
The Chinese University of Hong Kong
Shatin, Hong Kong
hchen@se.cuhk.edu.hk

**Jiajin Li**
Sauder School of Business
University of British Columbia
Vancouver, BC, Canada
jiajin.li@sauder.ubc.ca

**Anthony Man-Cho So**
SEEM
The Chinese University of Hong Kong
Shatin, Hong Kong
manchoso@se.cuhk.edu.hk

## Abstract

Solving bilevel optimization (BLO) problems to global optimality is generally intractable. A common surrogate is to compute a hyper-stationary point—a stationary point of the hyper-objective function obtained by minimizing or maximizing the upper-level objective over the lower-level solution set. Existing methods, however, either provide weak notions of stationarity or require restrictive assumptions to guarantee the smoothness of hyper-objective functions. In this paper, we eliminate these impractical assumptions and show that strong (Clarke) hyper-stationarity remains computable even when the hyper-objective is nonsmooth. Our key ingredient is a new structural property, called *set smoothness*, which captures the variational dependence of the lower-level solution set on the upper-level variable. We prove that this property holds for a broad class of BLO problems and ensures weak convexity (resp. concavity) of pessimistic (resp. optimistic) hyper-objective functions. Building on this foundation, we show that a zeroth-order algorithm that computes approximate Clarke hyper-stationary points with non-asymptotic convergence guarantees. To the best of our knowledge, this is the first computational guarantee for Clarke-type stationarity in nonsmooth BLO. Beyond this specific application, the set smoothness property emerges as a structural concept of independent interest, with potential to inform the analysis of broader classes of optimization and variational problems.

## 1 Introduction

Bilevel optimization (BLO) models hierarchical decision-making with two agents acting sequentially [13, 14]. The follower responds to the leader's decision by solving a lower-level optimization problem, while the leader seeks an optimal strategy to minimize its upper-level objective subject to this reaction. The follower's attitude plays a central role: If the follower is favorable (resp. adverse) to the leader, the resulting BLO is termed optimistic (resp. pessimistic) [13, 51, 36]. Formally, the optimistic and pessimistic BLO take the following forms:

39th Conference on Neural Information Processing Systems (NeurIPS 2025).

*Optimistic BLO:*

$$\begin{aligned} \min_{\boldsymbol{x}\in\mathbb{R}^m}\min_{\boldsymbol{y}\in\mathbb{R}^n} \quad & F(\boldsymbol{x},\boldsymbol{y}) \\ \text{subject to} \quad & \boldsymbol{y}\in\operatorname*{arg\,min}_{\boldsymbol{y}'\in\mathbb{R}^n} f(\boldsymbol{x},\boldsymbol{y}'), \end{aligned}$$

*Pessimistic BLO:*

$$\begin{aligned} \min_{\boldsymbol{x}\in\mathbb{R}^m}\max_{\boldsymbol{y}\in\mathbb{R}^n} \quad & F(\boldsymbol{x},\boldsymbol{y}) \\ \text{subject to} \quad & \boldsymbol{y}\in\operatorname*{arg\,min}_{\boldsymbol{y}'\in\mathbb{R}^n} f(\boldsymbol{x},\boldsymbol{y}'). \end{aligned}$$

These formulations appear in diverse domains such as Stackelberg games [14, 4, 50], hyperparameter optimization [17, 3, 6], reinforcement learning [29, 55, 21], and interdiction games [36, 5], among others. A standard approach to tackle such nested problems is to reformulate them into single-level problems via *hyper-objective functions*. Let $\mathcal{S}(\boldsymbol{x}) \coloneqq \operatorname*{arg\,min}_{\boldsymbol{y}'\in\mathbb{R}^n} f(\boldsymbol{x},\boldsymbol{y}')$ denote the follower's optimal response set. The *optimistic* and *pessimistic* hyper-objectives are then defined as

$$\varphi_o(\boldsymbol{x}) \coloneqq \min_{\boldsymbol{y}\in\mathcal{S}(\boldsymbol{x})} F(\boldsymbol{x},\boldsymbol{y}), \qquad \varphi_p(\boldsymbol{x}) \coloneqq \max_{\boldsymbol{y}\in\mathcal{S}(\boldsymbol{x})} F(\boldsymbol{x},\boldsymbol{y}). \tag{1}$$

Solving an optimistic (resp. pessimistic) BLO is therefore equivalent to minimizing the corresponding hyper-objective $\varphi_o$ (resp. $\varphi_p$).

Despite the single-level reformulation, the resulting hyper-objective functions are highly nonconvex [8, 31], which makes global optimization intractable. In practice, researchers therefore focus on finding stationary points rather than global minimizers, using algorithms such as implicit gradient descent [17, 18, 3] and fully first-order methods [32, 21]. These approaches assume that the lower-level problem is strongly convex, ensuring a unique solution, i.e., $\mathcal{S}(\boldsymbol{x}) \coloneqq \{\boldsymbol{y}^\star(\boldsymbol{x})\}$. Under this assumption, the hyper-objective reduces to a smooth function $\varphi(\boldsymbol{x}) \coloneqq F(\boldsymbol{x},\boldsymbol{y}^\star(\boldsymbol{x}))$ [19]. One can then seek an $\epsilon$-approximate hyper-stationary point satisfying $\|\nabla\varphi(\boldsymbol{x})\| \le \epsilon$. Convergence is well understood under smoothness and uniqueness assumptions [19], but these conditions rarely hold in practice. With multiple lower-level solutions, the existing methods break down.

To move beyond the singleton lower-level solution set, Kwon et al. [32] introduced a penalty-based framework that allows multiple follower solutions. Building on this idea, Chen et al. [8] obtained a refined scheme with near-optimal convergence. However, ensuring smoothness of the induced hyper-objective still demands strong regularity: The penalized model function $h_\sigma(\boldsymbol{x},\boldsymbol{y}) \coloneqq \sigma F(\boldsymbol{x},\boldsymbol{y}) + f(\boldsymbol{x},\boldsymbol{y})$ must satisfy, uniformly in $\sigma\in[0,\bar\sigma]$, an error bound or a Polyak–Łojasiewicz (PŁ) condition in $\boldsymbol{y}$. Such requirements are often unrealistic in practice, as $F$ and $f$ typically have mismatched structures. More fundamentally, the Kurdyka–Łojasiewicz (KŁ) exponent is not preserved under summation [23], so smoothness of hyper-objective functions cannot be guaranteed.

Without relying on these stringent conditions, Chen et al. [7] and Khanduri et al. [28] proposed algorithms for nonsmooth hyper-objectives; however, by their zero-respecting nature (cf. [8, Thm. 3.2]), they cannot in general approximate hyper-stationary points and thus only guarantee convergence to (approximate) Goldstein stationary points [20]—a relatively weak notion. By contrast, a separate line of work studies alternative stationarity concepts via reformulations [35, 52, 38, 1, 39]; yet these notions (e.g., KKT stationarity [37, Sec. 2.1] and penalization stationarity [54, Sec. 4.2]) are posed jointly in $(\boldsymbol{x},\boldsymbol{y})$ and do not ensure that, for a stationary pair $(\bar{\boldsymbol{x}},\bar{\boldsymbol{y}})$, the lower-level solution $\bar{\boldsymbol{y}}$ actually minimizes or maximizes $F(\bar{\boldsymbol{x}},\boldsymbol{y})$ over $\mathcal{S}(\bar{\boldsymbol{x}})$.

Given the above discussion, existing algorithms either fail to approximate a meaningful hyper-stationary point or rely on stringent assumptions to do so. This naturally leads to a fundamental question:

> **Can strong hyper-stationarity be computed in general settings**
> **where multiple lower-level solutions exist?**

Addressing this question is challenging for a simple reason chain. When the lower level admits multiple solutions, the induced hyper-objective is typically nonsmooth and, under standard assumptions, no better than Lipschitz continuous [7, Corollary 6.1]. At precisely this level of regularity, computing (stronger) approximate Clarke stationary points is, in general, computationally intractable [30, 46]. Thus Lipschitz regularity alone is too weak for algorithmic purposes, motivating new, verifiable structural conditions that make meaningful hyper-stationarity attainable.

**Our Contributions.** In this paper, we address the above challenges and show that (strong) Clarke stationarity of hyper-objective functions is computable for a broad class of BLO problems. As our key contribution, we identify a hidden weak convexity/concavity structure of the hyper-objective

in nonconvex–PŁ BLO,[1] which places the analysis within the well-studied weakly convex/concave framework. Within this setting, approximate hyper-stationarity admits a natural Clarke-subdifferential characterization that we leverage to obtain computable guarantees.

The foundation of our analysis is a new concept, *set smoothness* (Definition 3), which extends classical smoothness to set-valued mappings and encompasses several variational regularity notions [40, 15, 7, 27]. Building on this notion, we prove two complementary statements. First, if the lower-level solution mapping is set smooth, then the optimistic (resp. pessimistic) hyper-objective is weakly concave (resp. weakly convex). Second, a broad and verifiable condition guarantees set smoothness: When the lower-level function satisfies an error bound condition—equivalently, the PŁ condition—the solution mapping is set smooth. Together, these statements provide checkable criteria under which the hyper-objective inherits a weak convexity/concavity structure.

Once the hidden weak convexity/concavity of the hyper-objective is in place, approximate Clarke hyper-stationary points can be computed by a simple inexact zeroth-order scheme. In the weakly convex case, results based on the Moreau envelope [12, 56, 41] provide convergence and complexity guarantees. For the weakly concave case, however, no existing algorithmic guarantee is known, and the absence of a Moreau-type smoothing technique makes the analysis significantly more challenging. We overcome this by developing a novel convergence proof based on a Brøndsted–Rockafellar-type approximation result [43, Theorem 2], and establish, to the best of our knowledge, the first general computational guarantee for finding approximate Clarke stationary points of nonsmooth hyper-objective functions.

Overall, these developments, particularly set smoothness, provide a principled foundation for the computability of hyper-stationarity in BLO and open new avenues for other structured nonsmooth optimization problems.

**Organization.** This paper is organized as follows. Sec. 2 collects assumptions and preliminaries. Sec. 3 introduces set smoothness and uses it to reveal a weak convexity/concavity structure of the hyper-objective. Sec. 4 presents an inexact zeroth-order scheme and establishes convergence guarantees for computing approximate Clarke hyper-stationary points. Sec. 5 concludes with final remarks.

**Notation.** The notation used in this paper is mostly standard. We use $\|\boldsymbol{x}\|$ to denote the Euclidean norm of a vector $\boldsymbol{x}$ and $\|\mathbf{A}\|$ to denote the $l_2$ norm of a matrix $\mathbf{A}$. We use $\mathbb{B}(\boldsymbol{z}, r)$ to denote the ball centering at $\boldsymbol{z}$ with radius $r$, i.e., $\{\boldsymbol{x} : \|\boldsymbol{x} - \boldsymbol{z}\| \leq r\}$. For a scalar $\alpha \in \mathbb{R}$ and a set $\mathcal{S} \subseteq \mathbb{R}^n$, we use $\alpha \cdot \mathcal{S}$ to denote their product $\{\alpha \boldsymbol{x} : \boldsymbol{x} \in \mathcal{S}\}$. We define the distance from a vector $\boldsymbol{x} \in \mathbb{R}^n$ to $\mathcal{S}$ by $\mathrm{dist}(\boldsymbol{x}, \mathcal{S}) \coloneqq \min_{\boldsymbol{z} \in \mathcal{S}} \|\boldsymbol{x} - \boldsymbol{z}\|$ and the projection of $\boldsymbol{x}$ onto $\mathcal{S}$ by $\Pi_{\mathcal{S}}(\boldsymbol{x}) \coloneqq \arg\min_{\boldsymbol{z} \in \mathcal{S}} \|\boldsymbol{x} - \boldsymbol{z}\|$. We use $\mathrm{Conv}(\mathcal{S})$ to denote the convex hull of $\mathcal{S}$. For two sets $\mathcal{S}_1, \mathcal{S}_2 \subseteq \mathbb{R}^n$, define their Minkowski sum by $\mathcal{S}_1 + \mathcal{S}_2 \coloneqq \{\boldsymbol{x}_1 + \boldsymbol{x}_2 : \boldsymbol{x}_1 \in \mathcal{S}_1, \boldsymbol{x}_2 \in \mathcal{S}_2\}$, and define their Hausdorff distance (with respect to $\|\cdot\|$) by

$$d_{\mathrm{H}}(\mathcal{S}_1, \mathcal{S}_2) \coloneqq \max\left\{\sup_{\boldsymbol{x}_1 \in \mathcal{S}_1} \mathrm{dist}(\boldsymbol{x}_1, \mathcal{S}_2), \sup_{\boldsymbol{x}_2 \in \mathcal{S}_2} \mathrm{dist}(\boldsymbol{x}_2, \mathcal{S}_1)\right\}.$$

For a differentiable function $g : \mathbb{R}^m \times \mathbb{R}^n \to \mathbb{R}$, we use $\nabla g$ to denote its gradient w.r.t. the joint variables $(\boldsymbol{x}, \boldsymbol{y})$ and $\nabla_{\boldsymbol{x}} g$ (resp. $\nabla_{\boldsymbol{y}} g$) to denote its gradient w.r.t. $\boldsymbol{x}$ (resp. $\boldsymbol{y}$).

## 2 Preliminaries

In this paper, we focus on nonconvex-PŁ BLO problems and make the following assumptions:

**Assumption 1** (Lower-level Functions)**.**

(A1). *The function $f$ is $L_f$-smooth and twice differentiable. Moreover, $\nabla\nabla_{\boldsymbol{y}} f$ is $H_f$-Lipschitz continuous, i.e., for all $\boldsymbol{x}_1, \boldsymbol{x}_2 \in \mathbb{R}^m$ and $\boldsymbol{y}_1, \boldsymbol{y}_2 \in \mathbb{R}^n$,*

$$\|\nabla\nabla_{\boldsymbol{y}} f(\boldsymbol{x}_1, \boldsymbol{y}_1) - \nabla\nabla_{\boldsymbol{y}} f(\boldsymbol{x}_2, \boldsymbol{y}_2)\| \leq H_f (\|\boldsymbol{x}_1 - \boldsymbol{x}_2\| + \|\boldsymbol{y}_1 - \boldsymbol{y}_2\|).$$

(A2). *The solution set $\mathcal{S}(\boldsymbol{x}) = \arg\min_{\boldsymbol{y} \in \mathbb{R}^n} f(\boldsymbol{x}, \boldsymbol{y})$ is nonempty closed convex for all $\boldsymbol{x} \in \mathbb{R}^m$.*

---

[1]That is, BLO problems with a nonconvex upper-level objective and a lower-level function satisfying the PŁ condition.

(A3). *There exists a scalar $\tau > 0$ such that for all $\boldsymbol{x} \in \mathbb{R}^m$ and $\boldsymbol{y} \in \mathbb{R}^n$,*
$$\text{dist}(\boldsymbol{y}, \mathcal{S}(\boldsymbol{x})) \leq \tau \|\nabla_{\boldsymbol{y}} f(\boldsymbol{x}, \boldsymbol{y})\|.$$

**Assumption 2** (Upper-level and Hyper-objective Functions)**.**

(B1). *The function $F$ is $M_F$-Lipschitz continuous and $L_F$-smooth.*

(B2). *There exists $\boldsymbol{x}^\star \in \mathbb{R}^m$ such that $\varphi_o(\boldsymbol{x}^\star) > -\infty$ (resp. $\varphi_p(\boldsymbol{x}^\star) < +\infty$) for the optimistic (resp. pessimistic) setting.*

Assumptions (A1), (A2), and (B1) are standard in BLO settings; see, e.g., [21, 8, 19, 52, 1] and the references therein. Assumption (B2) guarantees that the hyper-objective functions are well-defined and is imposed without loss of generality. Assumption (A3) imposes an error bound in the lower-level variable that holds uniformly over the upper-level parameter. This requirement is strictly weaker than the strong convexity-in-$y$ conditions commonly used in prior work [18, 22, 21], as it allows the solution set $\arg\min_{\boldsymbol{y}} f(\boldsymbol{x}, \boldsymbol{y})$ to be multi-valued. Under $L_f$–smoothnes $f$ in $\boldsymbol{y}$, (A3) implies the PŁ inequality

$$f(\boldsymbol{x}, \boldsymbol{y}) - \min_{\boldsymbol{y} \in \mathbb{R}^n} f(\boldsymbol{x}, \boldsymbol{y}) \leq \frac{\tau L_f^2}{2} \|\nabla_{\boldsymbol{y}} f(\boldsymbol{x}, \boldsymbol{y})\|^2 \text{ for all } \boldsymbol{x} \in \mathbb{R}^m, \boldsymbol{y} \in \mathbb{R}^n,$$

vice versus [33, Theorem 3.1]. Hence, our setting aligns with the widely adopted nonconvex–PŁ framework for BLO [45, 52, 35].

Under the standing assumptions, we begin with the solution mapping $\mathcal{S}$, which under (A3) admits the following equivalent characterization:

$$\mathcal{S}(\boldsymbol{x}) = \{\boldsymbol{y} \in \mathbb{R}^n : \nabla_{\boldsymbol{y}} f(\boldsymbol{x}, \boldsymbol{y}) = \boldsymbol{0}\}. \tag{2}$$

Furthermore, under Assumption 1, the solution mapping $\mathcal{S}$ as well as the hyper-objective functions $\varphi_o$ and $\varphi_p$ are the Lipschitz continuous.

**Lemma 1** (Lipschitz Continuity of $\mathcal{S}(\boldsymbol{x})$)**.** *(cf. [7, Proposition 6.1]) Under Assumption 1, the lower-level solution set function is $M_{\mathcal{S}}$-Lipschitz continuous with $M_{\mathcal{S}} = L_f \tau$, i.e., for any $\boldsymbol{x}_1, \boldsymbol{x}_2 \in \mathbb{R}^m$,*
$$d_{\mathrm{H}}\left(\mathcal{S}(\boldsymbol{x}_1), \mathcal{S}(\boldsymbol{x}_2)\right) \leq M_{\mathcal{S}} \|\boldsymbol{x}_1 - \boldsymbol{x}_2\|.$$

**Lemma 2.** *(cf. [7, Proposition 5.3]) Suppose that Assumption 1 and 2 hold. For the optimistic (resp. pessimistic) setting, $\varphi_o$ (resp. $\varphi_p$) is $M_\varphi$-Lipschitz continuous with $M_\varphi = M_F(1 + L_f \tau)$.*

The Lipschitz continuity of the hyper-objective functions ensures that the Clarke and Goldstein subdifferentials are well defined.

**Definition 1** (Clarke Subdifferential)**.** *(cf. [9, Definition 1.1]) For a Lipschitz continuous function $g : \mathbb{R}^m \to \mathbb{R}$, the Clarke subdifferential of $g$ at a point $\boldsymbol{x} \in \mathbb{R}^m$ is defined by*
$$\partial g(\boldsymbol{x}) := \text{Conv}\left(\{\boldsymbol{s} \in \mathbb{R}^m : \exists\, \boldsymbol{x}' \to \boldsymbol{x},\ \nabla g(\boldsymbol{x}')\ \text{exists},\ \nabla g(\boldsymbol{x}') \to \boldsymbol{s}\}\right).$$
*We say that $\boldsymbol{x}$ is an $(\epsilon, \delta)$-approximate Clarke stationary point of $g$ if*

$$\text{dist}\left(\boldsymbol{0}, \bigcup_{\boldsymbol{z} \in \mathbb{B}(\boldsymbol{x}, \delta)} \partial g(\boldsymbol{z})\right) \leq \epsilon.$$

**Remark 1.** *The Clarke subdifferential $\partial g$ reduces to the gradient $\nabla g$ when $g$ is smooth. Moreover, if $g$ is convex, then $\partial g(\boldsymbol{x})$ coincides with the vanilla subgradients defined by $\{\boldsymbol{s} : g(\boldsymbol{z}) \geq g(\boldsymbol{x}) + \boldsymbol{s}^T(\boldsymbol{z} - \boldsymbol{x}) \,\forall\, \boldsymbol{z} \in \mathbb{R}^m\}$.*

Then, the Goldstein $\delta$-subdifferential at $\boldsymbol{x}$ can be constructed by the convex hull of the Clarke subdifferentials taken over a $\delta$-neighborhood of $\boldsymbol{x}$. Here is the formal definition of Goldstein $\delta$-subdifferential.

**Definition 2** (Goldstein $\delta$-Subdifferential)**.** *(cf. [20, Definition 2.2]) For a Lipschitz continuous function $g : \mathbb{R}^m \to \mathbb{R}$ and a scalar $\delta \geq 0$, the Goldstein $\delta$-subdifferential of $g$ at a point $\boldsymbol{x} \in \mathbb{R}^m$ is defined by*

$$\partial_\delta g(\boldsymbol{x}) := \text{Conv}\left(\left\{\bigcup_{\boldsymbol{z} \in \mathbb{B}(\boldsymbol{x}, \delta)} \partial g(\boldsymbol{z})\right\}\right).$$

*We say that $\boldsymbol{x}$ is an $(\epsilon, \delta)$-approximate Goldstein stationary point of $g$ if $\text{dist}(\boldsymbol{0}, \partial_\delta g(\boldsymbol{x})) \leq \epsilon$.*

Leveraging the Lipschitz continuity of the hyper-objective, recent work has established the computability of $(\epsilon, \delta)$–Goldstein hyper-stationary points [7]. However, Goldstein stationarity is strictly weaker and does not, in general, imply Clarke stationarity. Indeed, there exists a convex, 2-Lipschitz function $\tilde{g} : \mathbb{R}^2 \to \mathbb{R}$ and a point $\boldsymbol{x}$ such that $\boldsymbol{x}$ is $(0, \delta)$–Goldstein stationary while $\min_{\boldsymbol{z} \in \mathbb{B}(\boldsymbol{x}, \delta)} \operatorname{dist}(\boldsymbol{0}, \partial \tilde{g}(\boldsymbol{z})) \geq \frac{2}{\sqrt{5}}$; see [47, Proposition 2.7]. To obtain stronger algorithmic guarantees for hyper-objective minimization, we therefore focus on computing Clarke stationary points (and their $(\epsilon, \delta)$–approximate variants).

Despite the well-definedness, approximate Clarke stationarity is not achievable in finite time for general Lipschitz functions [30, 46]. A sufficient condition for its computability is the weak convexity of $g$ [12]. To elaborate on this, we review some basic properties of weakly convex functions. Given a function $g : \mathbb{R}^m \to \mathbb{R}$ and a scalar $r > 0$, we say that $g$ is $r$-*weakly convex* if the function $\boldsymbol{x} \mapsto g(\boldsymbol{x}) + \frac{r}{2}\|\boldsymbol{x}\|^2$ is convex. The following equivalent characterizations are useful for our analysis.

**Lemma 3** (Equivalent Characterizations of Weak Convexity). *(cf. [11, Theorem 3.1] and [2, Proposition 2.2]) For a Lipschitz continuous function $g : \mathbb{R}^m \to \mathbb{R}$, the following statements are equivalent:*

(i) *$g$ is $r$-weakly convex.*

(ii) *For any $\theta \in [0, 1]$ and $\boldsymbol{x}_1, \boldsymbol{x}_2 \in \mathbb{R}^m$, we have*

$$g\left(\theta \boldsymbol{x}_1 + (1 - \theta)\boldsymbol{x}_2\right) \leq \theta g(\boldsymbol{x}_1) + (1 - \theta)g(\boldsymbol{x}_2) + \frac{r}{2}\theta(1 - \theta)\|\boldsymbol{x}_1 - \boldsymbol{x}_2\|^2.$$

(iii) *For any $\boldsymbol{x}_1, \boldsymbol{x}_2 \in \mathbb{R}^m$ with $\partial g(\boldsymbol{x}_1) \neq \varnothing$, and all subgradients $\boldsymbol{v} \in \partial g(\boldsymbol{x}_1)$, we have*

$$\boldsymbol{v}^T(\boldsymbol{x}_2 - \boldsymbol{x}_1) \leq g(\boldsymbol{x}_2) - g(\boldsymbol{x}_1) + \frac{r}{2}\|\boldsymbol{x}_2 - \boldsymbol{x}_1\|^2.$$

For an $r$–weakly convex function $g : \mathbb{R}^m \to \mathbb{R}$ with $\gamma \in (0, \frac{1}{r})$, we define its Moreau envelope and the proximal mapping by

$$g_\gamma(\boldsymbol{x}) := \inf_{\boldsymbol{z} \in \mathbb{R}^n}\left\{g(\boldsymbol{z}) + \frac{1}{2\gamma}\|\boldsymbol{x} - \boldsymbol{z}\|^2\right\}, \qquad \operatorname{prox}_{\gamma, g}(\boldsymbol{x}) := \arg\min_{\boldsymbol{z} \in \mathbb{R}^n}\left\{g(\boldsymbol{z}) + \frac{1}{2\gamma}\|\boldsymbol{x} - \boldsymbol{z}\|^2\right\}.$$

Clearly, $\operatorname{prox}_{\gamma, g}(\boldsymbol{x})$ is single-valued and well-defined, when $g$ is $r$-weakly convex and $\gamma < \frac{1}{r}$. Next, we provide the standard result, which establishes a stationarity measure based on the gradient of the Moreau envelope.

**Lemma 4** (Properties of Moreau Envelope). *(cf. [56, Proposition 2.1]) Suppose that $g : \mathbb{R}^n \to \mathbb{R}$ is a $r$-weakly convex function and $\gamma < \frac{1}{r}$. The following hold:*

(i) $g_\gamma(\boldsymbol{x}) \leq g(\boldsymbol{x}) - \frac{1 - \gamma r}{2\gamma}\left\|\boldsymbol{x} - \operatorname{prox}_{\gamma, g}(\boldsymbol{x})\right\|^2.$

(ii) $\gamma \operatorname{dist}\left(\boldsymbol{0}, \partial g(\hat{\boldsymbol{x}})\right) \leq \left\|\boldsymbol{x} - \operatorname{prox}_{\gamma, g}(\boldsymbol{x})\right\| \leq \frac{\gamma}{1 - \gamma r}\operatorname{dist}\left(0, \partial g(\boldsymbol{x})\right).$

(iii) $\boldsymbol{x} = \operatorname{prox}_{\gamma, g}(\boldsymbol{x})$ *if and only if* $\boldsymbol{0} \in \partial g(\boldsymbol{x})$.

(iv) $\nabla g_\gamma(\boldsymbol{x}) = \frac{1}{\gamma}\left(\boldsymbol{x} - \operatorname{prox}_{\gamma, g}(\boldsymbol{x})\right).$

Lemma 4 (ii) and (iv) show that $\|\nabla g_\gamma(\boldsymbol{x})\|$ equals zero if and only if $\boldsymbol{x} = \operatorname{prox}_{\gamma, g}(\boldsymbol{x})$ and $\boldsymbol{0} \in \partial g(\boldsymbol{x})$. Thus $\|\nabla g_\gamma(\boldsymbol{x})\|$ is a valid Clarke stationarity measure. Moreover, by Lemma 4 (iv), if $\|\nabla g_\gamma(\boldsymbol{x})\| \leq \epsilon$, then $\|\operatorname{prox}_{\gamma, g}(\boldsymbol{x}) - \boldsymbol{x}\| = \gamma\|\nabla g_\gamma(\boldsymbol{x})\| \leq \gamma\epsilon$, hence $\operatorname{dist}\left(\boldsymbol{0}, \partial g(\operatorname{prox}_{\gamma, g}(\boldsymbol{x}))\right) \leq \epsilon$. Equivalently,

$$\|\nabla g_\gamma(\boldsymbol{x})\| \leq \epsilon \quad \Longrightarrow \quad \operatorname{dist}\left(\boldsymbol{0}, \bigcup_{\boldsymbol{z} \in \mathbb{B}(\boldsymbol{x}, \gamma\epsilon)} \partial g(\boldsymbol{z})\right) \leq \epsilon. \tag{3}$$

Since Davis and Drusvyatskiy [12] establish non-asymptotic rates for finding $\boldsymbol{x}$ with $\|\nabla g_\gamma(\boldsymbol{x})\| \leq \epsilon$ when $g$ is weakly convex, (3) implies that $(\epsilon, \gamma\epsilon)$–approximate Clarke stationarity is computable in

this regime. This observation motivates us to establish weak–convexity–type structure for hyper-objectives; see Sec. 3.

Before leaving this section, we record *weak concavity*, a notion closely related to weak convexity. For a function $g : \mathbb{R}^n \to \mathbb{R}$, we say that $g$ is $r$-weakly convex if $-g$ is $r$-weakly convex. We have the following facts.

**Fact 1.** *If a function $g : \mathbb{R}^m \to \mathbb{R}$ is $r$-weakly concave, then for any $\boldsymbol{x}_1, \boldsymbol{x}_2 \in \mathbb{R}^m$, and $\boldsymbol{v} \in \partial g(\boldsymbol{x}_1)$, we have $g(\boldsymbol{x}_2) \leq g(\boldsymbol{x}_1) + \boldsymbol{v}^T(\boldsymbol{x}_2 - \boldsymbol{x}_1) + \frac{r}{2}\|\boldsymbol{x}_2 - \boldsymbol{x}_1\|^2$.*

**Fact 2.** *If a function $g : \mathbb{R}^m \to \mathbb{R}$ is $r$-smooth, then $g$ is $r$-weakly convex and $r$-weakly concave, and the following inequality holds:*

$$|\theta g(\boldsymbol{x}_1) + (1 - \theta)g(\boldsymbol{x}_2) - g(\theta \boldsymbol{x}_1 + (1 - \theta)\boldsymbol{x}_2)| \leq \frac{r}{2}\theta(1 - \theta)\|\boldsymbol{x}_1 - \boldsymbol{x}_2\|^2.$$

## 3 Unveiling Hidden Structural Properties

This section is devoted to unveiling the hidden structural properties of the hyper-objective functions, which is the key contribution of this paper. Recall that the hyper-objective functions in (1) are defined by minimizing/maximizing the upper-level function w.r.t. $\boldsymbol{y}$ over the parameterized set $\mathcal{S}(\boldsymbol{x})$. We are motivated to investigate the property of the set-valued function $\mathcal{S}$. Inspired by the smoothness of real-valued functions, we propose a novel concept of smoothness for set-valued functions, formalized in Definition 3. As we will show, the lower-level solution set function $\mathcal{S}$ satisfies this smoothness property, which in turn ensures the weak concavity (resp. convexity) of $\varphi_o$ (resp. $\varphi_p$).

---

**Definition 3** (Set Smoothness). *For a set-valued function $\mathcal{Y} : \mathbb{R}^m \rightrightarrows \mathbb{R}^n$ with a convex domain $\mathrm{dom}(\mathcal{Y}) \subseteq \mathbb{R}^m$, we say that it is $L$-smooth if for any $\boldsymbol{x}_1, \boldsymbol{x}_2 \in \mathrm{dom}(\mathcal{Y})$, $\theta \in [0, 1]$, and all $\boldsymbol{y} \in \mathcal{Y}(\theta \boldsymbol{x}_1 + (1 - \theta)\boldsymbol{x}_2)$, there exist $\boldsymbol{y}_1 \in \mathcal{Y}(\boldsymbol{x}_1)$ and $\boldsymbol{y}_2 \in \mathcal{Y}(\boldsymbol{x}_2)$ such that*

$$\|\theta \boldsymbol{y}_1 + (1 - \theta)\boldsymbol{y}_2 - \boldsymbol{y}\| \leq \frac{L}{2}\theta(1 - \theta)\|\boldsymbol{x}_1 - \boldsymbol{x}_2\|^2; \tag{4}$$

$$\|\boldsymbol{y}_1 - \boldsymbol{y}_2\|^2 \leq L\|\boldsymbol{x}_1 - \boldsymbol{x}_2\|^2. \tag{5}$$

---

The condition (4) can be viewed as a natural extension of the gradient-Lipschitz smoothness condition for real-valued functions to the setting of set-valued mappings. It guarantees that a convex combination of $\boldsymbol{y}_1 \in \mathcal{Y}(\boldsymbol{x}_1)$ and $\boldsymbol{y}_2 \in \mathcal{Y}(\boldsymbol{x}_2)$ provides a close approximation to a point in $\mathcal{Y}(\theta \boldsymbol{x}_1 + (1 - \theta)\boldsymbol{x}_2)$, with an error that decays quadratically in $\|\boldsymbol{x}_1 - \boldsymbol{x}_2\|$. This yields the following set inclusion:

$$\mathcal{Y}(\theta \boldsymbol{x}_1 + (1 - \theta)\boldsymbol{x}_2) \subseteq \theta \mathcal{Y}(\boldsymbol{x}_1) + (1 - \theta)\mathcal{Y}(\boldsymbol{x}_2) + \frac{L}{2}\theta(1 - \theta)\|\boldsymbol{x}_1 - \boldsymbol{x}_2\|^2 \cdot \mathbb{B}(\boldsymbol{0}, 1). \tag{6}$$

Intuitively, (5) enforces a consistent branch selection between $\mathcal{Y}(\boldsymbol{x}_1)$ and $\mathcal{Y}(\boldsymbol{x}_2)$: The chosen representatives $\boldsymbol{y}_1$ and $\boldsymbol{y}_2$ must remain aligned (Lipschitz-close) as the input varies, thereby excluding cross-branch pairings that could make the interpolation in (4) hold trivially while the underlying geometry is severely mismatched.

**Example 1** (Why the condition (5) is needed: A trivialization for the condition (4)). Define the set-valued map $\mathcal{Y} : \mathbb{R} \rightrightarrows \mathbb{R}^2$ by $\mathcal{Y}(x) = \{(z, x) : z \in \mathbb{R}\}$. Pick $x_1 = a > 0$, $x_2 = -a$, and $\theta = \frac{1}{2}$; then $\theta x_1 + (1 - \theta)x_2 = 0$ and $\mathcal{Y}(0) = \{(z, 0) : z \in \mathbb{R}\}$. Choose $\boldsymbol{y} = \boldsymbol{0} \in \mathcal{Y}(0)$, $\boldsymbol{y}_1 = (K, a) \in \mathcal{Y}(a)$, and $\boldsymbol{y}_2 = (-K, -a) \in \mathcal{Y}(-a)$. We have

$$\tfrac{1}{2}\boldsymbol{y}_1 + \tfrac{1}{2}\boldsymbol{y}_2 = \boldsymbol{y},$$

so the condition (4) holds with zero error even though $\|\boldsymbol{y}_1 - \boldsymbol{y}_2\| = 2\sqrt{a^2 + K^2}$ can be made arbitrarily large as $K \to \infty$. Hence the condition (4) alone does not preclude severely mismatched pairings on a convex domain. In contrast, (5) enforces $\|\boldsymbol{y}_1 - \boldsymbol{y}_2\|^2 \leq L\|x_1 - x_2\|^2 = 4La^2$, which forces $K^2 \leq (L - 1)a^2$ and thereby rules out such cross-branch selections unless the Lipschitz modulus is correspondingly large. □

With the notion of set smoothness in place, we now present our first main theoretical result. It shows that set smoothness serves as the key vehicle for establishing weak convexity/concavity of parametric optimization problems with coupled constraints: Under mild Lipschitz-type assumptions, the induced value function inherits weak convexity (or weak concavity). This is formalized in Theorem 1 below.

**Theorem 1** (Implication of Set Smoothness). *Consider a real-valued function $g : \mathbb{R}^m \times \mathbb{R}^n \to \mathbb{R}$ and a set-valued function $\mathcal{Y} : \mathbb{R}^m \rightrightarrows \mathbb{R}^n$. Let $\phi(\boldsymbol{x}) := \max_{\boldsymbol{y} \in \mathcal{Y}(\boldsymbol{x})} g(\boldsymbol{x}, \boldsymbol{y})$ and $\mathcal{D} := \{\boldsymbol{x} : \phi(\boldsymbol{x}) > -\infty\}$. Assume that $\mathcal{D}$ is a nonempty closed convex set, $\mathcal{Y}$ is $L_{\mathcal{Y}}$-smooth on $\mathcal{D}$, and $g$ is $M_g$-Lipschitz continuous w.r.t. $\boldsymbol{y}$, $L_g$-smooth on $\mathcal{D} \times \mathrm{Conv}\left(\bigcup_{\boldsymbol{x} \in \mathcal{D}} \mathcal{Y}(\boldsymbol{x})\right)$. Then, the function $\phi$ is $\rho$-weakly convex with $\rho = M_g L_{\mathcal{Y}} + L_g(1 + L_{\mathcal{Y}})$.*

We now instantiate the framework in the bilevel setting. Our first step is to certify *set smoothness* for the lower-level solution map. Under the error-bound (EB) condition in Assumption 1 (A3), the mapping $\mathcal{S} : \boldsymbol{x} \mapsto \arg\min_{\boldsymbol{y} \in \mathbb{R}^n} f(\boldsymbol{x}, \boldsymbol{y})$ is $L_{\mathcal{S}}$-smooth (Theorem 2). Combining this with Theorem 1 shows that the pessimistic hyper-objective $\varphi_p$ inherits weak convexity (resp. the optimistic $\varphi_o$ inherits weak concavity).

**Theorem 2** (EB Implies Set Smoothness). *If a function $f : \mathbb{R}^m \times \mathbb{R}^n \to \mathbb{R}$ satisfies Assumption 1, then its associated solution set function $\mathcal{S} : \boldsymbol{x} \mapsto \arg\min_{\boldsymbol{y} \in \mathbb{R}^n} f(\boldsymbol{x}, \boldsymbol{y})$ is $L_{\mathcal{S}}$-smooth with $L_{\mathcal{S}} = \max\{2H_f \tau(1 + 9L_f^2 \tau^2), 4L_f^2 \tau^2\}$.*

**Proof idea (why residual backfilling is essential)** Fix $\boldsymbol{x}_1, \boldsymbol{x}_2$ and $\theta \in (0, 1)$, and set $\boldsymbol{x}^\theta := \theta \boldsymbol{x}_1 + (1 - \theta)\boldsymbol{x}_2$. Given any $\boldsymbol{y} \in \mathcal{S}(\boldsymbol{x}^\theta)$, our goal is to select $\boldsymbol{y}_1 \in \mathcal{S}(\boldsymbol{x}_1)$ and $\boldsymbol{y}_2 \in \mathcal{S}(\boldsymbol{x}_2)$ so that (4) and (5) hold. A natural choice is to project $\boldsymbol{y}$ onto the endpoint fibers, yielding $\bar{\boldsymbol{y}}_i := \Pi_{\mathcal{S}(\boldsymbol{x}_i)}(\boldsymbol{y})$ for $i = 1, 2$. Using Lemma 1, it is easy to see that this naive selection satisfies (5).

However, even when each fiber $\mathcal{S}(\boldsymbol{x})$ is convex, the midpoint $\bar{\boldsymbol{y}}^\theta := \theta \bar{\boldsymbol{y}}_1 + (1 - \theta)\bar{\boldsymbol{y}}_2$ may correspond, at $\boldsymbol{x}^\theta$, to a *different local selection* of the set-valued map $\mathcal{S}(\cdot)$ than the given $\boldsymbol{y} \in \mathcal{S}(\boldsymbol{x}^\theta)$. Consequently, in general multi-solution settings the naive midpoint error can be *first-order*,

$$\|\bar{\boldsymbol{y}}^\theta - \boldsymbol{y}\| = \Theta(\|\boldsymbol{x}_1 - \boldsymbol{x}_2\|),$$

which motivates an additional correction to *synchronize the selections*.

We therefore *align the midpoint* and *backfill the residual*: First project the naive midpoint to the middle fiber, $\hat{\boldsymbol{y}} := \Pi_{\mathcal{S}(\boldsymbol{x}^\theta)}(\bar{\boldsymbol{y}}^\theta)$, and then use the residual $\boldsymbol{y} - \hat{\boldsymbol{y}}$ to refine the endpoint representatives:

$$\boxed{\boldsymbol{y}_i := \Pi_{\mathcal{S}(\boldsymbol{x}_i)}\big(\bar{\boldsymbol{y}}_i + (\boldsymbol{y} - \hat{\boldsymbol{y}})\big), \qquad i = 1, 2.}$$

This construction cancels the first-order branch mismatch in the convex combination and leaves only a quadratic remainder. Consequently, (4) holds while (5) remains valid.

**Remark 2.** *Think of $\mathcal{S}(\boldsymbol{x})$ as a family of convex "fibers". The direct projections $\bar{\boldsymbol{y}}_1, \bar{\boldsymbol{y}}_2$ may live on selections that are not synchronized with the selection containing $\boldsymbol{y}$, so their convex combination carries a first-order drift. Projecting $\bar{\boldsymbol{y}}^\theta$ to $\mathcal{S}(\boldsymbol{x}^\theta)$ identifies the correct selection at the midpoint; adding the* same *residual $\boldsymbol{y} - \hat{\boldsymbol{y}}$ to both endpoints moves them to the same selection as $\boldsymbol{y}$, making the first-order terms cancel in the average and exposing the desired $\mathcal{O}(\|\boldsymbol{x}_1 - \boldsymbol{x}_2\|^2)$ behavior.*

Theorem 2 is not limited to the lower-level problem of BLO but applies to general parametric optimization problems. The established set smoothness property offers new insights into the structure of the solution mapping, which goes beyond the variational conditions considered in the literature [40, 27, 15, 7, 53].

**Remark 3** (Local Conditions are Sufficient). *Suppose the solution mapping $\mathcal{S}$ is defined on a bounded convex domain $\mathcal{D} \subseteq \mathbb{R}^m$. To ensure Theorem 2, it suffices that Assumption 1 holds on the set $\mathcal{D} \times \mathcal{Y}$, where*

$$\mathcal{Y} = \mathrm{Conv}\left(\bigcup_{\boldsymbol{x} \in \mathcal{D}} \mathcal{S}(\boldsymbol{x})\right) + \tfrac{1}{2} M_{\mathcal{S}} \, \mathrm{diam}(\mathcal{D}) \, \mathbb{B}(\boldsymbol{0}, 1).$$

The following simple example shows that the set smoothness of $\mathcal{S}$ does not, in general, require Assumption 1. This suggests that alternative sufficient conditions may guarantee set smoothness; identifying such conditions is an interesting direction for future work.

**Example 2.** Consider $f : \mathbb{R}^2 \to \mathbb{R}$ defined by $f(x, y) = g(\sin x + y)$, where $g : \mathbb{R} \to \mathbb{R}$ has a nonempty set of minimizers $\mathcal{V} = \arg\min_{z \in \mathbb{R}} g(z)$. Then the solution set admits a closed form:

$$\mathcal{S}(x) = \arg\min_{y \in \mathbb{R}} f(x, y) = \mathcal{V} - \sin x := \{v - \sin x : v \in \mathcal{V}\}.$$

In particular, $\mathcal{S}$ is 1-smooth in the sense of Definition 3 (since it is a translation of the fixed set $\mathcal{V}$ by the scalar $-\sin x$), even though $f$ need not satisfy Assumption 1. $\quad\square$

With Theorems 1 and 2 in place, we now state our main result on the weak convexity/concavity of the hyper-objective $\varphi_o$ (resp. $\varphi_p$).

**Theorem 3** (Weak convexity/concavity of the hyper-objectives). *Assume Assumptions 1 and 2. Let $L_{\mathcal{S}}$ be the set-smoothness modulus of $\mathcal{S}$ from Theorem 2, and define $\rho := M_F L_{\mathcal{S}} + L_F (1 + L_{\mathcal{S}})$. Then the following hold:*

(i) *The optimistic hyper-objective $\varphi_o$ is $\rho$-weakly concave.*

(ii) *The pessimistic hyper-objective $\varphi_p$ is $\rho$-weakly convex.*

*Proof of Theorem 3.* Theorem 2 guarantees that the set-valued function $\mathcal{S}$ is $L_{\mathcal{S}}$-smooth. Then, the result (ii) directly follows from Theorem 1. Hence, we only need to prove (i). Note that

$$-\varphi_o(\boldsymbol{x}) = - \min_{\boldsymbol{y} \in \mathcal{S}(\boldsymbol{x})} F(\boldsymbol{x}, \boldsymbol{y}) = \max_{\boldsymbol{y} \in \mathcal{S}(\boldsymbol{x})} -F(\boldsymbol{x}, \boldsymbol{y}).$$

We see that $-\varphi_o$ is $\rho$-weakly convex by Theorem 1. It follows that $\varphi_o$ is $\rho$-weakly concave. $\quad\square$

Theorem 3 establishes the weak concavity/convexity of the hyper-objectives in nonconvex–PŁ bilevel optimization (BLO). This stands in contrast to classical results (e.g., [19, Lemma 2.2]), which impose strong convexity of the lower level to obtain smooth hyper-objectives. Our result is significant because it places the minimization of these generally *nonsmooth* hyper-objectives within the framework of weakly concave/convex optimization. As a consequence, computing approximate Clarke hyper-stationary points becomes tractable—an avenue we pursue in the next section. Crucially, all of these developments hinge on the *set smoothness* property (Definition 3), highlighting the utility of this notion.

**Remark 4** (Lower-level Constraints Matter). *Under Assumptions 1 and 2, imposing an upper-level constraint $\boldsymbol{x} \in \mathcal{X} \subseteq \mathbb{R}^m$ with $\mathcal{X}$ nonempty, closed, and convex preserves the conclusions of Theorem 3: The functions $\varphi_o(\boldsymbol{x}) + \iota_{\mathcal{X}}(\boldsymbol{x})$ and $\varphi_p(\boldsymbol{x}) + \iota_{\mathcal{X}}(\boldsymbol{x})$ remain weakly concave and weakly convex, respectively, where $\iota_{\mathcal{X}}$ denotes the indicator of $\mathcal{X}$. In contrast, adding a lower-level constraint $\boldsymbol{y} \in \mathcal{Y}$ can destroy the weak concavity/convexity of the hyper-objectives, because the set smoothness of $\mathcal{S}$ may fail in this case; see Example 3. Developing structural conditions that recover such properties for lower-level constrained BLO is an interesting direction for future work.*

**Example 3.** Let $\mathcal{Y} = [0, 1] \times [0, 1]$. Consider the pessimistic bilevel problem *with a lower-level constraint*:

$$\min_{x \in \mathbb{R}} \max_{\boldsymbol{y} \in \mathbb{R}^2} \quad -\mathbf{1}^\top \boldsymbol{y} \tag{7}$$
$$\text{s.t.} \quad \boldsymbol{y} \in \arg\min_{\boldsymbol{y}' \in \mathcal{Y}} \|\boldsymbol{y}' - (x, 2)\|^2.$$

Assumptions 1 and 2 are directly satisfied for (7), except *for the unconstrained lower level*; the only deviation here is the added constraint $\boldsymbol{y} \in \mathcal{Y}$.

The lower-level solution set is the projection of $(x, 2)$ onto the box $\mathcal{Y}$, hence

$$\mathcal{S}(x) = \begin{cases} \{(0, 1)\}, & x \leq 0, \\ \{(x, 1)\}, & 0 \leq x \leq 1, \\ \{(1, 1)\}, & x \geq 1. \end{cases}$$

Therefore the pessimistic hyper-objective is

$$\varphi_p(x) = \begin{cases} -1, & x \leq 0, \\ -x - 1, & 0 \leq x \leq 1, \\ -2, & x \geq 1. \end{cases}$$

This function is *not* weakly convex. Indeed, for any $\rho \geq 0$ consider $h_\rho(x) := \varphi_p(x) + \frac{\rho}{2}x^2$. Then $h_\rho$ has left and right derivatives at $x = 0$ given by $h'_\rho(0^-) = 0$ and $h'_\rho(0^+) = -1$ (the quadratic term has zero slope at 0), which violates the monotonicity of one-sided derivatives required by convexity. Hence no $\rho$ makes $h_\rho$ convex, i.e., $\varphi_p$ is not weakly convex. $\quad\square$

# 4 Computing Approximate Clarke Hyper-stationarity

Equipped with the weak convexity/concavity of the hyper-objectives, our next goal is to establish the computability of Clarke stationary points. First-order methods are impractical here because subgradients of the hyper-objectives are typically unavailable. In contrast, under mild conditions—e.g., $F(\boldsymbol{x}, \cdot)$ is concave (resp. convex) so that the inner maximization (resp. minimization) is tractable, the *function values* of the hyper-objectives can be (approximately) evaluated at a given $\boldsymbol{x}$ [16, 44]. This motivates the use of *zeroth-order* methods for minimizing hyper-objectives [7, 34]. In particular, we adopt the inexact zeroth-order method (IZOM) in Algorithm 1, which employs a deterministic subroutine $\mathcal{A}$ to approximately evaluate $\varphi_\beta(\boldsymbol{x})$ (with additive accuracy $w$) by solving the inner problem in (1); see [7, 24–26] for practical implementations of $\mathcal{A}$.

---

**Algorithm 1** Inexact Zeroth-order Method (cf. [7, Algorithm 2])

---

**Input:** Radius $\varepsilon > 0$, iteration number $T \in \mathbb{N}$, stepsize $\eta$, initial point $\boldsymbol{x}_0 \in \mathbb{R}^m$, inexact error $w > 0$, and mode parameter $\beta \in \{1, 0\}$

    **for** $t = 0, 1, \ldots, T-1$ **do**

        Sample $\boldsymbol{u}_t$ from the the uniform distribution on the unit sphere in $\mathbb{R}^m$

        Compute $\mathcal{A}_w^\beta(\boldsymbol{x}_t + \varepsilon \boldsymbol{u}_t)$ and $\mathcal{A}_w^\beta(\boldsymbol{x}_t - \varepsilon \boldsymbol{u}_t)$ by subroutine $\mathcal{A}$

        Set $\tilde{G}(\boldsymbol{x}_t) = \frac{m}{2\varepsilon}(\mathcal{A}_w^\beta(\boldsymbol{x}_t + \varepsilon \boldsymbol{u}_t) - \mathcal{A}_w^\beta(\boldsymbol{x}_t - \varepsilon \boldsymbol{u}_t))\boldsymbol{u}_t$

        $\boldsymbol{x}_{t+1} = \boldsymbol{x}_t - \eta \tilde{G}(\boldsymbol{x}_t)$

    **end for**

**Output:** $\bar{\boldsymbol{x}}$ uniformly chosen from $\{\boldsymbol{x}_t\}_{t=0}^{T-1}$

---

---

**Algorithm 2** Deterministic Subroutine $\mathcal{A}$

---

**Input:** Accuracy $w > 0$, iterate point $\boldsymbol{x} \in \mathbb{R}^m$, and mode $\beta \in \{1, 0\}$

    **if** $\beta = 1$ **then**

        Compute a value $\tilde{\varphi}(\boldsymbol{x})$ satisfying $|\tilde{\varphi}(\boldsymbol{x}) - \varphi_o(\boldsymbol{x})| \leq w$

    **else**

        Compute a value $\tilde{\varphi}(\boldsymbol{x})$ satisfying $|\tilde{\varphi}(\boldsymbol{x}) - \varphi_p(\boldsymbol{x})| \leq w$

    **end if**

**Output:** $\mathcal{A}_w^\beta(\boldsymbol{x}) = \tilde{\varphi}(\boldsymbol{x})$

---

Let $\varphi_{p,\gamma}$ denote the Moreau envelope of $\varphi_p$ with parameter $\gamma$. We quantify hyper-stationarity as follows: In the optimistic case we use the approximate Clarke stationarity measure, i.e., Definition 1, while in the pessimistic case we use the gradient norm of the envelope, i.e., $\|\nabla \varphi_{p,\gamma}(\boldsymbol{x})\|$. These two criteria can be unified in principle via (3); in either form they are strictly stronger than the Goldstein stationarity measure; see Sec. 2.

We then present the main theorem of this section.

**Theorem 4.** *Suppose that Assumptions 1 and 2 hold. Given an iteration number $T \in \mathbb{N}$, set $\eta = \Theta(m^{-\frac{1}{2}}T^{-\frac{1}{2}}), \varepsilon = \mathcal{O}(T^{-\frac{1}{2}}), w = \mathcal{O}(m^{-\frac{3}{4}}T^{-\frac{3}{4}})$ for Algorithm 1. Then, the following hold:*

  (i) *Let $\Delta_o := \varphi_o(\boldsymbol{x}_0) - \min_{\boldsymbol{x}} \varphi_o(\boldsymbol{x}) + 2M_\varphi \varepsilon$ with $M_\varphi$ given in Lemma 2. For optimistic BLO, we have*

$$\mathbb{E}\left[\text{dist}\left(\boldsymbol{0}, \bigcup_{\boldsymbol{z} \in \mathbb{B}(\bar{\boldsymbol{x}}, \delta)} \partial \varphi_o(\boldsymbol{z})\right)^2\right] = \mathcal{O}\left(\frac{\sqrt{m}(\Delta_o + 1)}{\sqrt{T}}\right) \text{ with } \delta = \mathcal{O}\left(T^{-\frac{1}{4}}\right).$$

  (ii) *Let $\gamma \in (0, \frac{1}{\rho+1})$ with $\rho > 0$ given in Theorem 3, and $\Delta_p := \varphi_{p,\gamma}(\boldsymbol{x}_0) - \min_{\boldsymbol{x}} \varphi_{p,\gamma}(\boldsymbol{x})$. For pessimistic BLO, we have*

$$\mathbb{E}[\|\nabla \varphi_{p,\gamma}(\bar{\boldsymbol{x}})\|^2] = \mathcal{O}\left(\frac{\sqrt{m}(\Delta_p + 1)}{\sqrt{T}}\right).$$

Theorem 4 demonstrates, for the first time, that approximate Clarke hyper-stationarity is computable for nonconvex-PŁ BLO in both optimistic and pessimistic settings. This result significantly improves the existing computational guarantees for nonsmooth hyper-objective functions, which are mainly based on the Goldstein stationarity [7, 28]. The proof of the optimistic case relies on a Brøndsted-Rockafellar-like relation, details of which can be found in Appendix D.1.

# 5 Conclusion and Discussion

In this paper, we established the first theoretical guarantee for computing approximate Clarke hyper-stationarity in nonconvex-PŁ BLO. The key step is unveiling the hidden structural properties of hyper-objective functions via the newly introduced smoothness concept for set-valued functions. Specifically, we proved that (i) the smoothness of the set-valued function $\mathcal{Y}$ ensures the weak convexity of the function $\boldsymbol{x} \mapsto \max_{\boldsymbol{y} \in \mathcal{Y}(\boldsymbol{x})} \phi(\boldsymbol{x}, \boldsymbol{y})$; and (ii) the lower-level solution set function of BLO satisfies set smoothness. Consequently, we obtained the weak convexity/concavity of hyper-objective functions. With these properties in hand, we showed that an inexact zeroth-order method can compute approximate Clarke stationary points of hyper-objective functions.

We believe that our developments contribute to a deeper understanding of the computability properties of BLO and open up several directions for future research. First, with the established structural properties, our work calls for designing faster algorithms for computing Clarke hyper-stationarity. Second, it would be valuable to generalize our methodology to establish adapted properties for BLO in other settings (e.g., structured lower-level constrained BLO [28]). Furthermore, our set smoothness property, along with Theorem 1, may find applications in other fields such as coupled minmax optimization [48] and set-valued optimization [27], where set-valued functions play a central role.

## Ackonwledgements

Jiajin Li was supported by a Natural Sciences and Engineering Research Council of Canada Discovery Grant RGPIN-2025-05817. Anthony Man-Cho So was supported in part by the Hong Kong Research Grants Council (RGC) General Research Fund (GRF) project CUHK 14204823.

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

# A  Missing Proofs for Sec. 2

## A.1  Proof of Lemma 1

*Proof.* Thanks to Assumption 1 (A3), we have

$$\text{dist}(\boldsymbol{y}_1, \mathcal{S}(\boldsymbol{x}_2)) \leq \tau \|\nabla_{\boldsymbol{y}} f(\boldsymbol{x}_2, \boldsymbol{y}_1)\|,$$

for any $\boldsymbol{y}_1 \in \mathcal{S}(\boldsymbol{x}_1)$. Moreover, we have

$$\|\nabla_{\boldsymbol{y}} f(\boldsymbol{x}_2, \boldsymbol{y}_1)\| = \|\nabla_{\boldsymbol{y}} f(\boldsymbol{x}_2, \boldsymbol{y}_1) - \nabla_{\boldsymbol{y}} f(\boldsymbol{x}_1, \boldsymbol{y}_1)\| \leq L_f \|\boldsymbol{x}_1 - \boldsymbol{x}_2\|,$$

where the equality follows from $\nabla_{\boldsymbol{y}} f(\boldsymbol{x}_1, \boldsymbol{y}_1) = \boldsymbol{0}$, and the inequality is due to the $L_f$-smoothness of $f$. Putting them together yields $\text{dist}(\boldsymbol{y}_1, \mathcal{S}(\boldsymbol{x}_2)) \leq L_f \tau \|\boldsymbol{x}_1 - \boldsymbol{x}_2\|$ for all $\boldsymbol{y}_1 \in S(\boldsymbol{x}_1)$.

By the same argument with $\boldsymbol{x}_1$ and $\boldsymbol{x}_2$ interchanged, we have $\text{dist}(\boldsymbol{y}_2, \mathcal{S}(\boldsymbol{x}_1)) \leq L_f \tau \|\boldsymbol{x}_1 - \boldsymbol{x}_2\|$ for all $\boldsymbol{y}_2 \in S(\boldsymbol{x}_1)$. By the definition of $d_{\text{H}}(\mathcal{S}(\boldsymbol{x}_1), \mathcal{S}(\boldsymbol{x}_2))$, we conclude

$$d_{\text{H}}(\mathcal{S}(\boldsymbol{x}_1), \mathcal{S}(\boldsymbol{x}_2)) = \max \left\{ \sup_{\boldsymbol{y}_1 \in \mathcal{S}(\boldsymbol{x}_1)} \text{dist}(\boldsymbol{y}_1, \mathcal{S}(\boldsymbol{x}_2)), \sup_{\boldsymbol{y}_2 \in \mathcal{S}(\boldsymbol{x}_2)} \text{dist}(\boldsymbol{y}_2, \mathcal{S}(\boldsymbol{x}_1)) \right\}$$
$$\leq L_f \tau \|\boldsymbol{x}_1 - \boldsymbol{x}_2\|.$$

This completes the proof. □

## A.2  Proof of Lemma 2

*Proof.* We prove the $M_\varphi$-Lipschitz continuity for $\varphi_p$ only, as the argument for $\varphi_o$ is entirely analogous.

Then, it suffices to show that for any $\boldsymbol{x}_1, \boldsymbol{x}_2 \in \text{dom}(\varphi_p)$,

$$|\varphi_p(\boldsymbol{x}_1) - \varphi_p(\boldsymbol{x}_2)| \leq M_\varphi \|\boldsymbol{x}_2 - \boldsymbol{x}_2\|.$$

Note that $\mathcal{S}(\boldsymbol{x})$ is closed but not necessarily compact. We can only find a sequence $\{\boldsymbol{y}_1^k\}_{k \in \mathbb{N}} \subseteq \mathcal{S}(\boldsymbol{x}_1)$ such that $F(\boldsymbol{x}_1, \boldsymbol{y}_1^k) \to \sup_{\boldsymbol{y} \in \mathcal{S}(\boldsymbol{x}_1)} F(\boldsymbol{x}_1, \boldsymbol{y}) = \varphi_p(\boldsymbol{x}_1)$. Let $\boldsymbol{y}_2^k := \Pi_{\mathcal{S}(\boldsymbol{x}_2)}(\boldsymbol{y}_1^k)$. Then, by $\boldsymbol{y}_2^k \in S(\boldsymbol{x}_2)$, we have $\varphi_p(\boldsymbol{x}_2) = \sup_{\boldsymbol{y} \in \mathcal{S}(\boldsymbol{x}_2)} F(\boldsymbol{x}_2, \boldsymbol{y}) \geq F(\boldsymbol{x}_2, \boldsymbol{y}_2^k)$. This observation, combined with the $M_F$-Lipschitz continuity of $F$, yields

$$F(\boldsymbol{x}_1, \boldsymbol{y}_1^k) - \varphi_p(\boldsymbol{x}_2) \leq F(\boldsymbol{x}_1, \boldsymbol{y}_1^k) - F(\boldsymbol{x}_2, \boldsymbol{y}_2^k) \leq M_F \left( \|\boldsymbol{x}_1 - \boldsymbol{x}_2\| + \|\boldsymbol{y}_1^k - \boldsymbol{y}_2^k\| \right),$$

where the last inequality follows from Assumption 2 (B1).

Moreover, we have

$$\|\boldsymbol{y}_1^k - \boldsymbol{y}_2^k\| = \|\boldsymbol{y}_1^k - \Pi_{\mathcal{S}(\boldsymbol{x}_2)}(\boldsymbol{y}_1^k)\| = \text{dist}(\boldsymbol{y}_1^k, \mathcal{S}(\boldsymbol{x}_2)) \leq d_{\text{H}}(\mathcal{S}(\boldsymbol{x}_1), \mathcal{S}(\boldsymbol{x}_2)) \leq M_{\mathcal{S}} \|\boldsymbol{x}_1 - \boldsymbol{x}_2\|,$$

where the last inequality is due to Lemma 1. Then, we can see that for all $k \in \mathbb{N}$,

$$F(\boldsymbol{x}_1, \boldsymbol{y}_1^k) - \varphi_p(\boldsymbol{x}_2) \leq M_F(1 + M_{\mathcal{S}}) \|\boldsymbol{x}_1 - \boldsymbol{x}_2\|.$$

Letting $k \to \infty$ and recalling $F(\boldsymbol{x}_1, \boldsymbol{y}_1^k) \leftarrow \varphi_p(\boldsymbol{x}_1)$, we obtain

$$\varphi_p(\boldsymbol{x}_1) - \varphi_p(\boldsymbol{x}_2) \leq M_F(1 + M_{\mathcal{S}}) \|\boldsymbol{x}_1 - \boldsymbol{x}_2\|,$$

which completes the proof. □

## A.3  Proof of Fact 1

*Proof.* By [49, Proposition 4.4], the weakly convex function $-g$ is locally Lipschitz (hence so is $g$), so the Clarke subdifferential $\partial g$ is well defined. Moreover, the Clarke subdifferential satisfies $\partial(-g) = -\partial g$ [10, Prop. 2.3.1]. Applying Lemma 3 (iii) to $-g$ then yields the claim. □

# B  Proof of Theorem 1

*Proof.* To establish the weak convexity of $\phi$, we verify the condition in Lemma 3 (ii). Specifically, we show that there exists a constant $\rho > 0$ such that, for all $\theta \in [0,1]$ and all $\boldsymbol{x}_1, \boldsymbol{x}_2 \in \mathcal{D}$,

$$\phi(\boldsymbol{x}^\theta) \leq \theta\phi(\boldsymbol{x}_1) + (1-\theta)\phi(\boldsymbol{x}_2) + \frac{\rho}{2}\theta(1-\theta)\|\boldsymbol{x}_1 - \boldsymbol{x}_2\|^2, \tag{8}$$

where we let $\boldsymbol{x}^\theta := \theta\boldsymbol{x}_1 + (1-\theta)\boldsymbol{x}_2$ for notation convenience.

By definition, $\phi(\boldsymbol{x}^\theta) = \max_{\boldsymbol{y}' \in \mathcal{Y}(\boldsymbol{x}^\theta)} g(\boldsymbol{x}^\theta, \boldsymbol{y}')$. Hence (8) will follow if we show that, for all $\boldsymbol{y} \in \mathcal{Y}(\boldsymbol{x}^\theta)$,

$$g(\boldsymbol{x}^\theta, \boldsymbol{y}) - \theta\,\phi(\boldsymbol{x}_1) - (1-\theta)\,\phi(\boldsymbol{x}_2) \leq \tfrac{\rho}{2}\theta(1-\theta)\,\|\boldsymbol{x}_1 - \boldsymbol{x}_2\|^2. \tag{9}$$

Taking the maximization over $\boldsymbol{y} \in \mathcal{Y}(\boldsymbol{x}^\theta)$ then yields (8).

We proceed via the $L_{\mathcal{Y}}$-smoothness of $\mathcal{Y}$, which guarantees the existence of $\boldsymbol{y}_1 \in \mathcal{Y}(\boldsymbol{x}_1)$ and $\boldsymbol{y}_2 \in \mathcal{Y}(\boldsymbol{x}_2)$ such that

$$\|\theta\boldsymbol{y}_1 + (1-\theta)\boldsymbol{y}_2 - \boldsymbol{y}\| \leq \frac{L_{\mathcal{Y}}}{2}\theta(1-\theta)\|\boldsymbol{x}_1 - \boldsymbol{x}_2\|^2; \tag{10}$$

$$\|\boldsymbol{y}_1 - \boldsymbol{y}_2\|^2 \leq L_{\mathcal{Y}}\|\boldsymbol{x}_1 - \boldsymbol{x}_2\|^2. \tag{11}$$

Then, using the fact that $\phi(\boldsymbol{x}_i) = \max_{\boldsymbol{y}' \in \mathcal{Y}(\boldsymbol{x}_i)} g(\boldsymbol{x}_i, \boldsymbol{y}') \geq g(\boldsymbol{x}_i, \boldsymbol{y}_i)$ for $i = 1, 2$, we have

$$
\begin{aligned}
&g\left(\boldsymbol{x}^\theta, \boldsymbol{y}\right) - \theta\phi(\boldsymbol{x}_1) - (1-\theta)\phi(\boldsymbol{x}_2) \\
\leq& g\left(\boldsymbol{x}^\theta, \boldsymbol{y}\right) - \theta g(\boldsymbol{x}_1, \boldsymbol{y}_1) - (1-\theta)g(\boldsymbol{x}_2, \boldsymbol{y}_2) \\
\leq& g\left(\boldsymbol{x}^\theta, \boldsymbol{y}\right) - g\left(\boldsymbol{x}^\theta, \theta\boldsymbol{y}_1 + (1-\theta)\boldsymbol{y}_2\right) + \frac{L_g}{2}\theta(1-\theta)\left(\|\boldsymbol{x}_1 - \boldsymbol{x}_2\|^2 + \|\boldsymbol{y}_1 - \boldsymbol{y}_2\|^2\right),
\end{aligned}
$$

where the last inequality is due to the $L_g$-smoothness of $g$ on $\mathcal{D} \times \text{Conv}(\bigcup_{\boldsymbol{x} \in \mathcal{D}} \mathcal{Y}(\boldsymbol{x}))$ and Fact 2.

This, together with the $M_g$-Lipschitz continuity of $g$ w.r.t. $\boldsymbol{y}$ and the set-smoothness inequalities (10) and (11), yields

$$
\begin{aligned}
&g\left(\boldsymbol{x}^\theta, \boldsymbol{y}\right) - \theta\phi(\boldsymbol{x}_1) - (1-\theta)\phi(\boldsymbol{x}_2) \\
\leq& M_g\|\theta\boldsymbol{y}_1 + (1-\theta)\boldsymbol{y}_2 - \boldsymbol{y}\| + \frac{L_g}{2}\theta(1-\theta)\left(\|\boldsymbol{x}_1 - \boldsymbol{x}_2\|^2 + \|\boldsymbol{y}_1 - \boldsymbol{y}_2\|^2\right) \\
\leq& \frac{M_g L_{\mathcal{Y}}}{2}\theta(1-\theta)\|\boldsymbol{x}_1 - \boldsymbol{x}_2\|^2 + \frac{L_g}{2}\theta(1-\theta)\left(\|\boldsymbol{x}_1 - \boldsymbol{x}_2\|^2 + L_{\mathcal{Y}}\|\boldsymbol{x}_1 - \boldsymbol{x}_2\|^2\right) \\
=& \frac{M_g L_{\mathcal{Y}} + L_g(1 + L_{\mathcal{Y}})}{2}\theta(1-\theta)\|\boldsymbol{x}_1 - \boldsymbol{x}_2\|^2,
\end{aligned}
$$

for all $\boldsymbol{y} \in \mathcal{Y}(\boldsymbol{x}^\theta)$. We prove the desired inequality (9) with $\rho = M_g L_{\mathcal{Y}} + L_g(1 + L_{\mathcal{Y}})$.

$\square$

# C  Proof of Theorem 2

We start by stating a lemma that will be used in the sequel.

**Lemma 5.** *Under Assumption 1 (A1), for any $\boldsymbol{x}_1, \boldsymbol{x}_2 \in \mathbb{R}^m$ and $\boldsymbol{y}_1, \boldsymbol{y}_2 \in \mathbb{R}^n$, we have*

(i) $\quad \|\nabla_{\boldsymbol{y}} f(\boldsymbol{x}_1, \boldsymbol{y}_1) - \nabla_{\boldsymbol{y}} f(\boldsymbol{x}_2, \boldsymbol{y}_2) - \nabla\nabla_{\boldsymbol{y}} f(\boldsymbol{x}_2, \boldsymbol{y}_2)(\boldsymbol{x}_1 - \boldsymbol{x}_2, \boldsymbol{y}_1 - \boldsymbol{y}_2)\|$

$\qquad \leq \dfrac{H_f}{2}\left(\|\boldsymbol{x}_1 - \boldsymbol{x}_2\|^2 + \|\boldsymbol{y}_1 - \boldsymbol{y}_2\|^2\right);$

(ii) $\quad \|\theta\nabla_{\boldsymbol{y}} f(\boldsymbol{x}_1, \boldsymbol{y}_1) + (1-\theta)\nabla_{\boldsymbol{y}} f(\boldsymbol{x}_2, \boldsymbol{y}_2) - \nabla_{\boldsymbol{y}} f(\boldsymbol{x}^\theta, \boldsymbol{y}^\theta)\|$

$\qquad \leq \dfrac{H_f}{2}\theta(1-\theta)\left(\|\boldsymbol{x}_1 - \boldsymbol{x}_2\|^2 + \|\boldsymbol{y}_1 - \boldsymbol{y}_2\|^2\right), \quad \forall\, \theta \in [0,1],$

*where $\boldsymbol{x}^\theta := \theta\boldsymbol{x}_1 + (1-\theta)\boldsymbol{x}_2$ and $\boldsymbol{y}^\theta := \theta\boldsymbol{y}_1 + (1-\theta)\boldsymbol{y}_2$.*

*Proof of Lemma 5.* (i) The argument directly follows from [42, Lemma 1]:

$$\left\| \nabla_{\boldsymbol{y}} f(\boldsymbol{x}_1, \boldsymbol{y}_1) - \nabla_{\boldsymbol{y}} f(\boldsymbol{x}_2, \boldsymbol{y}_2) - \nabla \nabla_{\boldsymbol{y}} f(\boldsymbol{x}_2, \boldsymbol{y}_2)(\boldsymbol{x}_1 - \boldsymbol{x}_2, \boldsymbol{y}_1 - \boldsymbol{y}_2) \right\|$$

$$= \left\| \int_0^1 \nabla \nabla_{\boldsymbol{y}} f(\boldsymbol{x}_1 + t(\boldsymbol{x}_2 - \boldsymbol{x}_1), \boldsymbol{y}_1 + t(\boldsymbol{y}_2 - \boldsymbol{y}_1))(\boldsymbol{x}_1 - \boldsymbol{x}_2, \boldsymbol{y}_1 - \boldsymbol{y}_2)\, \mathrm{d}t \right.$$
$$\left. - \nabla \nabla_{\boldsymbol{y}} f(\boldsymbol{x}_2, \boldsymbol{y}_2)(\boldsymbol{x}_1 - \boldsymbol{x}_2, \boldsymbol{y}_1 - \boldsymbol{y}_2) \right\|$$

$$= \left\| \int_0^1 \left( \nabla \nabla_{\boldsymbol{y}} f(\boldsymbol{x}_1 + t(\boldsymbol{x}_2 - \boldsymbol{x}_1), \boldsymbol{y}_1 + t(\boldsymbol{y}_2 - \boldsymbol{y}_1)) - \nabla \nabla_{\boldsymbol{y}} f(\boldsymbol{x}_2, \boldsymbol{y}_2) \right)(\boldsymbol{x}_1 - \boldsymbol{x}_2, \boldsymbol{y}_1 - \boldsymbol{y}_2)\, \mathrm{d}t \right\|$$

$$\leq \int_0^1 \left\| \left( \nabla \nabla_{\boldsymbol{y}} f(\boldsymbol{x}_1 + t(\boldsymbol{x}_2 - \boldsymbol{x}_1), \boldsymbol{y}_1 + t(\boldsymbol{y}_2 - \boldsymbol{y}_1)) - \nabla \nabla_{\boldsymbol{y}} f(\boldsymbol{x}_2, \boldsymbol{y}_2) \right)(\boldsymbol{x}_1 - \boldsymbol{x}_2, \boldsymbol{y}_1 - \boldsymbol{y}_2) \right\|\, \mathrm{d}t$$

$$\leq \int_0^1 \left\| \nabla \nabla_{\boldsymbol{y}} f(\boldsymbol{x}_1 + t(\boldsymbol{x}_2 - \boldsymbol{x}_1), \boldsymbol{y}_1 + t(\boldsymbol{y}_2 - \boldsymbol{y}_1)) - \nabla \nabla_{\boldsymbol{y}} f(\boldsymbol{x}_2, \boldsymbol{y}_2) \right\| \left\| (\boldsymbol{x}_1 - \boldsymbol{x}_2, \boldsymbol{y}_1 - \boldsymbol{y}_2) \right\|\, \mathrm{d}t$$

$$\leq \int_0^1 H_f t \left\| (\boldsymbol{x}_1 - \boldsymbol{x}_2, \boldsymbol{y}_1 - \boldsymbol{y}_2) \right\| \cdot \left\| (\boldsymbol{x}_1 - \boldsymbol{x}_2, \boldsymbol{y}_1 - \boldsymbol{y}_2) \right\|\, \mathrm{d}t$$

$$= \frac{H_f}{2} \left( \|\boldsymbol{x}_1 - \boldsymbol{x}_2\|^2 + \|\boldsymbol{y}_1 - \boldsymbol{y}_2\|^2 \right),$$

where the first inequality is due to Jensen's inequality; the second inequality is due to the Cauchy inequality; and the third inequality uses the $H_f$-Lipschitz continuity of $\nabla \nabla_{\boldsymbol{y}} f$.

(ii) Using the result of (i), we have

$$\left\| \nabla_{\boldsymbol{y}} f(\boldsymbol{x}_1, \boldsymbol{y}_1) - \nabla_{\boldsymbol{y}} f(\boldsymbol{x}^\theta, \boldsymbol{y}^\theta) - \nabla \nabla_{\boldsymbol{y}} f(\boldsymbol{x}^\theta, \boldsymbol{y}^\theta)(\boldsymbol{x}_1 - \boldsymbol{x}^\theta, \boldsymbol{y}_1 - \boldsymbol{y}^\theta) \right\|$$
$$\leq \frac{H_f}{2} \left( \|\boldsymbol{x}_1 - \boldsymbol{x}^\theta\|^2 + \|\boldsymbol{y}_1 - \boldsymbol{y}^\theta\|^2 \right).$$

It follows from $\boldsymbol{x}^\theta = \theta \boldsymbol{x}_1 + (1 - \theta)\boldsymbol{x}_2$ and $\boldsymbol{y}^\theta = \theta \boldsymbol{y}_1 + (1 - \theta)\boldsymbol{y}_2$ that

$$\left\| \nabla_{\boldsymbol{y}} f(\boldsymbol{x}_1, \boldsymbol{y}_1) - \nabla_{\boldsymbol{y}} f(\boldsymbol{x}^\theta, \boldsymbol{y}^\theta) - (1 - \theta)\nabla \nabla_{\boldsymbol{y}} f(\boldsymbol{x}^\theta, \boldsymbol{y}^\theta)(\boldsymbol{x}_1 - \boldsymbol{x}_2, \boldsymbol{y}_1 - \boldsymbol{y}_2) \right\|$$
$$\leq \frac{H_f}{2}(1 - \theta)^2 \left( \|\boldsymbol{x}_1 - \boldsymbol{x}_2\|^2 + \|\boldsymbol{y}_1 - \boldsymbol{y}_2\|^2 \right). \tag{12}$$

Using the same arguments with $(\boldsymbol{x}_1, \boldsymbol{y}_1)$ replaced by $(\boldsymbol{x}_2, \boldsymbol{y}_2)$, we have

$$\left\| \nabla_{\boldsymbol{y}} f(\boldsymbol{x}_2, \boldsymbol{y}_2) - \nabla_{\boldsymbol{y}} f(\boldsymbol{x}^\theta, \boldsymbol{y}^\theta) - \theta \nabla \nabla_{\boldsymbol{y}} f(\boldsymbol{x}^\theta, \boldsymbol{y}^\theta)(\boldsymbol{x}_2 - \boldsymbol{x}_1, \boldsymbol{y}_2 - \boldsymbol{y}_1) \right\|$$
$$\leq \frac{H_f}{2}\theta^2 \left( \|\boldsymbol{x}_1 - \boldsymbol{x}_2\|^2 + \|\boldsymbol{y}_1 - \boldsymbol{y}_2\|^2 \right). \tag{13}$$

Then, the desired inequality follows from the weighted sum $\theta \times (12) + (1 - \theta) \times (13)$, and the triangle inequality that

$$\theta \left\| \nabla_{\boldsymbol{y}} f(\boldsymbol{x}_1, \boldsymbol{y}_1) - \nabla_{\boldsymbol{y}} f(\boldsymbol{x}^\theta, \boldsymbol{y}^\theta) - (1 - \theta)\nabla \nabla_{\boldsymbol{y}} f(\boldsymbol{x}^\theta, \boldsymbol{y}^\theta)(\boldsymbol{x}_1 - \boldsymbol{x}_2, \boldsymbol{y}_1 - \boldsymbol{y}_2) \right\|$$
$$+ (1 - \theta) \left\| \nabla_{\boldsymbol{y}} f(\boldsymbol{x}_2, \boldsymbol{y}_2) - \nabla_{\boldsymbol{y}} f(\boldsymbol{x}^\theta, \boldsymbol{y}^\theta) - \theta \nabla \nabla_{\boldsymbol{y}} f(\boldsymbol{x}^\theta, \boldsymbol{y}^\theta)(\boldsymbol{x}_2 - \boldsymbol{x}_1, \boldsymbol{y}_2 - \boldsymbol{y}_1) \right\|$$
$$\geq \left\| \theta \nabla_{\boldsymbol{y}} f(\boldsymbol{x}_1, \boldsymbol{y}_1) + (1 - \theta)\nabla_{\boldsymbol{y}} f(\boldsymbol{x}_2, \boldsymbol{y}_2) - \nabla_{\boldsymbol{y}} f(\boldsymbol{x}^\theta, \boldsymbol{y}^\theta) \right\|.$$

$$\square$$

We are now ready to prove Proposition 2. Fix $\theta \in [0, 1]$ and $\boldsymbol{x}_1, \boldsymbol{x}_2 \in \mathbb{R}^m$, and let $\boldsymbol{y} \in \mathcal{S}(\theta \boldsymbol{x}_1 + (1 - \theta)\boldsymbol{x}_2)$. Our goal is to construct $\boldsymbol{y}_1 \in \mathcal{S}(\boldsymbol{x}_1)$ and $\boldsymbol{y}_2 \in \mathcal{S}(\boldsymbol{x}_2)$ such that (4) and (5) hold.

## C.1  Step 1: Choose projection points as the candidate approximation points.

Let $\bar{\boldsymbol{y}}_1 := \Pi_{\mathcal{S}(\boldsymbol{x}_1)}(\boldsymbol{y})$ and $\bar{\boldsymbol{y}}_2 := \Pi_{\mathcal{S}(\boldsymbol{x}_2)}(\boldsymbol{y})$. For simplicity, we write $\boldsymbol{x}^\theta = \theta \boldsymbol{x}_1 + (1 - \theta)\boldsymbol{x}_2$ and $\bar{\boldsymbol{y}}^\theta = \theta \bar{\boldsymbol{y}}_1 + (1 - \theta)\bar{\boldsymbol{y}}_2$. We start with giving basic estimates on $\|\bar{\boldsymbol{y}}_1 - \bar{\boldsymbol{y}}_2\|$ and $\mathrm{dist}(\bar{\boldsymbol{y}}^\theta, \mathcal{S}(\boldsymbol{x}^\theta))$, in Claim 1.

**Claim 1.** *Let $L_0 := H_f \tau (1 + M_{\mathcal{S}}^2)/2$. The following hold:*

$$\|\bar{\boldsymbol{y}}_1 - \bar{\boldsymbol{y}}_2\| \leq M_{\mathcal{S}}\|\boldsymbol{x}_1 - \boldsymbol{x}_2\| \tag{14}$$

$$\mathrm{dist}\left(\bar{\boldsymbol{y}}^\theta, \mathcal{S}\left(\boldsymbol{x}^\theta\right)\right) \leq L_0 \theta(1 - \theta)\|\boldsymbol{x}_1 - \boldsymbol{x}_2\|^2. \tag{15}$$

*Proof of Claim 1.* We have

$$
\begin{aligned}
\|\bar{\boldsymbol{y}}_1 - \bar{\boldsymbol{y}}_2\| &\leq \|\bar{\boldsymbol{y}}_1 - \boldsymbol{y}\| + \|\boldsymbol{y} - \bar{\boldsymbol{y}}_2\| \\
&= \operatorname{dist}(\boldsymbol{y}, \mathcal{S}(\boldsymbol{x}_1)) + \operatorname{dist}(\boldsymbol{y}, \mathcal{S}(\boldsymbol{x}_2)) \\
&\leq d_{\mathrm{H}}\left(\mathcal{S}\left(\boldsymbol{x}^\theta\right), \mathcal{S}(\boldsymbol{x}_1)\right) + d_{\mathrm{H}}\left(\mathcal{S}\left(\boldsymbol{x}^\theta\right), \mathcal{S}(\boldsymbol{x}_2)\right) \\
&\leq M_{\mathcal{S}}\|\boldsymbol{x}_1 - (\theta\boldsymbol{x}_1 + (1-\theta)\boldsymbol{x}_2)\| + M_{\mathcal{S}}\|\boldsymbol{x}_2 - (\theta\boldsymbol{x}_1 + (1-\theta)\boldsymbol{x}_2)\| \\
&= M_{\mathcal{S}}\|\boldsymbol{x}_1 - \boldsymbol{x}_2\|,
\end{aligned}
$$

where the second inequality follows from $\boldsymbol{y} \in \mathcal{S}(\boldsymbol{x}^\theta)$ and the definition of the Hausdorff distance; the third inequality is due to Lemma 1 and $\boldsymbol{x}^\theta = \theta\boldsymbol{x}_1 + (1-\theta)\boldsymbol{x}_2$. This proves (14).

We then prove (15). First, by the $H_f$-Lipschitz continuity of $\nabla\nabla_{\boldsymbol{y}} f$ and Lemma 5 (ii), we have

$$
\begin{aligned}
&\left\|\theta\nabla_{\boldsymbol{y}} f(\boldsymbol{x}_1, \bar{\boldsymbol{y}}_1) + (1-\theta)\nabla_{\boldsymbol{y}} f(\boldsymbol{x}_2, \bar{\boldsymbol{y}}_2) - \nabla_{\boldsymbol{y}} f(\boldsymbol{x}^\theta, \bar{\boldsymbol{y}}^\theta)\right\| \\
&\leq \frac{H_f}{2}\theta(1-\theta)\left(\|\boldsymbol{x}_1 - \boldsymbol{x}_2\|^2 + \|\bar{\boldsymbol{y}}_1 - \bar{\boldsymbol{y}}_2\|^2\right) \\
&\leq \frac{H_f}{2}\theta(1-\theta)\left(1 + M_{\mathcal{S}}^2\right)\|\boldsymbol{x}_1 - \boldsymbol{x}_2\|^2,
\end{aligned}
\tag{16}
$$

where the second inequality is due to (14).

Since $\bar{\boldsymbol{y}}_1 \in \mathcal{S}(\boldsymbol{x}_1)$ and $\bar{\boldsymbol{y}}_2 \in \mathcal{S}(\boldsymbol{x}_2)$, we have the first-order optimality

$$
\nabla_{\boldsymbol{y}} f(\boldsymbol{x}_1, \bar{\boldsymbol{y}}_1) = 0, \qquad \nabla_{\boldsymbol{y}} f(\boldsymbol{x}_2, \bar{\boldsymbol{y}}_2) = 0.
$$

Substituting these identities into (16) yields

$$
\left\|\nabla_{\boldsymbol{y}} f\left(\boldsymbol{x}^\theta, \bar{\boldsymbol{y}}^\theta\right)\right\| \leq \frac{H_f}{2}\theta(1-\theta)\left(1 + M_{\mathcal{S}}^2\right)\|\boldsymbol{x}_1 - \boldsymbol{x}_2\|^2.
$$

Combining the error bound from Assumption 1, $\operatorname{dist}(\bar{\boldsymbol{y}}^\theta, \mathcal{S}(\boldsymbol{x}^\theta)) \leq \tau\|\nabla_{\boldsymbol{y}} f(\boldsymbol{x}^\theta, \bar{\boldsymbol{y}}^\theta)\|$, with the preceding inequality, we obtain

$$
\operatorname{dist}\left(\bar{\boldsymbol{y}}^\theta, \mathcal{S}\left(\boldsymbol{x}^\theta\right)\right) \leq \frac{H_f\tau}{2}\theta(1-\theta)\left(1 + M_{\mathcal{S}}^2\right)\|\boldsymbol{x}_1 - \boldsymbol{x}_2\|^2,
$$

which accords with (15) with $L_0 = H_f\tau(1 + M_{\mathcal{S}}^2)/2$. Claim 1 is proved. $\qquad\square$

Note that (15) controls only the distance from $\bar{\boldsymbol{y}}^\theta$ to the set $\mathcal{S}(\boldsymbol{x}^\theta)$; it does not guarantee that the specific point $\theta\bar{\boldsymbol{y}}_1 + (1-\theta)\bar{\boldsymbol{y}}_2$ lies on the same branch as the metric projection $\Pi_{\mathcal{S}(\boldsymbol{x}^\theta)}(\bar{\boldsymbol{y}}^\theta)$. Hence one cannot conclude that $\|\theta\bar{\boldsymbol{y}}_1 + (1-\theta)\bar{\boldsymbol{y}}_2 - \boldsymbol{y}\| = \mathcal{O}(\|\boldsymbol{x}_1 - \boldsymbol{x}_2\|^2)$ in general. Therefore we cannot simply take $\boldsymbol{y}_1 = \bar{\boldsymbol{y}}_1$ and $\boldsymbol{y}_2 = \bar{\boldsymbol{y}}_2$, which motivates the rectification in Step 2.

### C.2 Step 2: Translate the candidate approximation points

Let $\hat{\boldsymbol{y}} := \Pi_{\mathcal{S}(\boldsymbol{x}^\theta)}(\bar{\boldsymbol{y}}^\theta)$, $\hat{\boldsymbol{y}}_1 := \bar{\boldsymbol{y}}_1 + (\boldsymbol{y} - \hat{\boldsymbol{y}})$, and $\hat{\boldsymbol{y}}_2 := \bar{\boldsymbol{y}}_2 + (\boldsymbol{y} - \hat{\boldsymbol{y}})$. We will bound the three quantities $\|\hat{\boldsymbol{y}} - \boldsymbol{y}\|$, $\operatorname{dist}(\hat{\boldsymbol{y}}_1, \mathcal{S}(\boldsymbol{x}_1))$, and $\operatorname{dist}(\hat{\boldsymbol{y}}_2, \mathcal{S}(\boldsymbol{x}_2))$.

**Claim 2.** *Let* $L_1 := H_f\tau\left(1 + 17M_{\mathcal{S}}^2\right)/2$. *The following hold:*

$$
\|\hat{\boldsymbol{y}} - \boldsymbol{y}\| \leq 2\theta(1-\theta)M_{\mathcal{S}}\|\boldsymbol{x}_1 - \boldsymbol{x}_2\|;
\tag{17}
$$

$$
\operatorname{dist}\left(\hat{\boldsymbol{y}}_1, \mathcal{S}(\boldsymbol{x}_1)\right) \leq L_1(1-\theta)^2\|\boldsymbol{x}_1 - \boldsymbol{x}_2\|^2; \qquad \operatorname{dist}\left(\hat{\boldsymbol{y}}_2, \mathcal{S}(\boldsymbol{x}_2)\right) \leq L_1\theta^2\|\boldsymbol{x}_1 - \boldsymbol{x}_2\|^2.
\tag{18}
$$

*Proof of Claim 2.* We begin with the proof of (17). The non-expansiveness of the projection operator $\Pi_{\mathcal{S}(\boldsymbol{x}^\theta)}(\cdot)$ yields

$$
\|\hat{\boldsymbol{y}} - \boldsymbol{y}\| = \left\|\Pi_{\mathcal{S}(\boldsymbol{x}^\theta)}(\bar{\boldsymbol{y}}^\theta) - \Pi_{\mathcal{S}(\boldsymbol{x}^\theta)}(\boldsymbol{y})\right\| \leq \left\|\bar{\boldsymbol{y}}^\theta - \boldsymbol{y}\right\| \leq \theta\left\|\bar{\boldsymbol{y}}_1 - \boldsymbol{y}\right\| + (1-\theta)\left\|\bar{\boldsymbol{y}}_2 - \boldsymbol{y}\right\|.
\tag{19}
$$

Moreover, by $\bar{\boldsymbol{y}}_i = \Pi_{\mathcal{S}(\boldsymbol{x}_i)}(\boldsymbol{y})$ for $i = 1, 2$, $\boldsymbol{y} \in \mathcal{S}(\boldsymbol{x}^\theta)$, and the $M_{\mathcal{S}}$-Lipschitz continuity of $\mathcal{S}$ from Lemma 1, we have

$$
\|\bar{\boldsymbol{y}}_i - \boldsymbol{y}\| = \operatorname{dist}(\boldsymbol{y}, \mathcal{S}(\boldsymbol{x}_i)) \leq d_{\mathrm{H}}\left(\mathcal{S}\left(\boldsymbol{x}^\theta\right), \mathcal{S}(\boldsymbol{x}_i)\right) \leq M_{\mathcal{S}}\left\|\boldsymbol{x}_i - \boldsymbol{x}^\theta\right\|, \quad i = 1, 2.
\tag{20}
$$

Combining (19) and (20) leads to

$$\|\hat{\boldsymbol{y}} - \boldsymbol{y}\| \le \theta M_{\mathcal{S}} \|\boldsymbol{x}_1 - \boldsymbol{x}^{\theta}\| + (1-\theta)M_{\mathcal{S}} \|\boldsymbol{x}_2 - \boldsymbol{x}^{\theta}\| = 2\theta(1-\theta)M_{\mathcal{S}} \|\boldsymbol{x}_1 - \boldsymbol{x}_2\|.$$

We then continue to prove (18). It suffices to upper bound $\|\nabla_{\boldsymbol{y}} f(\boldsymbol{x}_1, \hat{\boldsymbol{y}}_1)\|$, due to Assumption 1 (A3). By Lemma 5, we have $\nabla_{\boldsymbol{y}} f(\boldsymbol{x}_1, \hat{\boldsymbol{y}}_1)$:

$$\|\nabla_{\boldsymbol{y}} f(\boldsymbol{x}_1, \hat{\boldsymbol{y}}_1) - \nabla_{\boldsymbol{y}} f(\boldsymbol{x}_1, \bar{\boldsymbol{y}}_1) - \nabla\nabla_{\boldsymbol{y}} f(\boldsymbol{x}_1, \bar{\boldsymbol{y}}_1)(\boldsymbol{0}, \hat{\boldsymbol{y}}_1 - \bar{\boldsymbol{y}}_1)\| \le \frac{H_f}{2}\|\hat{\boldsymbol{y}}_1 - \bar{\boldsymbol{y}}_1\|^2.$$

Noticing $\bar{\boldsymbol{y}}_1 \in \mathcal{S}(\boldsymbol{x}_1)$, we have $\nabla_{\boldsymbol{y}} f(\boldsymbol{x}_1, \bar{\boldsymbol{y}}_1) = \boldsymbol{0}$, and thus

$$\|\nabla_{\boldsymbol{y}} f(\boldsymbol{x}_1, \hat{\boldsymbol{y}}_1) - \nabla\nabla_{\boldsymbol{y}} f(\boldsymbol{x}_1, \bar{\boldsymbol{y}}_1)(\boldsymbol{0}, \hat{\boldsymbol{y}}_1 - \bar{\boldsymbol{y}}_1)\| \le \frac{H_f}{2}\|\hat{\boldsymbol{y}}_1 - \bar{\boldsymbol{y}}_1\|^2.$$

Using the triangle inequality and the identity $\hat{\boldsymbol{y}}_1 - \bar{\boldsymbol{y}}_1 = \boldsymbol{y} - \hat{\boldsymbol{y}}$ (by the definition of $\hat{\boldsymbol{y}}_1$), we obtain

$$\|\nabla_{\boldsymbol{y}} f(\boldsymbol{x}_1, \hat{\boldsymbol{y}}_1)\| \le \|\nabla\nabla_{\boldsymbol{y}} f(\boldsymbol{x}_1, \bar{\boldsymbol{y}}_1)(\boldsymbol{0}, \boldsymbol{y} - \hat{\boldsymbol{y}})\| + \frac{H_f}{2}\|\boldsymbol{y} - \hat{\boldsymbol{y}}\|^2. \tag{21}$$

We proceed to control $\|\nabla\nabla_{\boldsymbol{y}} f(\boldsymbol{x}_1, \bar{\boldsymbol{y}}_1)(\boldsymbol{0}, \boldsymbol{y} - \hat{\boldsymbol{y}})\|$. To do so, we first estimate a closely related norm $\|\nabla\nabla_{\boldsymbol{y}} f(\boldsymbol{x}^{\theta}, \boldsymbol{y})(\boldsymbol{0}, \boldsymbol{y} - \hat{\boldsymbol{y}})\|$. We apply Lemma 5 again to obtain

$$\left\|\nabla_{\boldsymbol{y}} f\left(\boldsymbol{x}^{\theta}, \hat{\boldsymbol{y}}\right) - \nabla_{\boldsymbol{y}} f\left(\boldsymbol{x}^{\theta}, \boldsymbol{y}\right) - \nabla\nabla_{\boldsymbol{y}} f\left(\boldsymbol{x}^{\theta}, \boldsymbol{y}\right)(\boldsymbol{0}, \hat{\boldsymbol{y}} - \boldsymbol{y})\right\| \le \frac{H_f}{2}\|\hat{\boldsymbol{y}} - \boldsymbol{y}\|^2.$$

Note that both $\boldsymbol{y}$ and $\hat{\boldsymbol{y}}$ belong to the set $\mathcal{S}(\boldsymbol{x}^{\theta})$, which leads to

$$\nabla_{\boldsymbol{y}} f\left(\boldsymbol{x}^{\theta}, \hat{\boldsymbol{y}}\right) = \nabla_{\boldsymbol{y}} f\left(\boldsymbol{x}^{\theta}, \boldsymbol{y}\right) = \boldsymbol{0}.$$

It follows that

$$\left\|\nabla\nabla_{\boldsymbol{y}} f\left(\boldsymbol{x}^{\theta}, \boldsymbol{y}\right)(\boldsymbol{0}, \boldsymbol{y} - \hat{\boldsymbol{y}})\right\| \le \frac{H_f}{2}\|\boldsymbol{y} - \hat{\boldsymbol{y}}\|^2. \tag{22}$$

Putting everything together yields

$$\begin{aligned}
&\|\nabla_{\boldsymbol{y}} f(\boldsymbol{x}_1, \hat{\boldsymbol{y}}_1)\| \\
&\le \left\|\left(\nabla\nabla_{\boldsymbol{y}} f(\boldsymbol{x}_1, \bar{\boldsymbol{y}}_1) - \nabla\nabla_{\boldsymbol{y}} f\left(\boldsymbol{x}^{\theta}, \boldsymbol{y}\right)\right)(\boldsymbol{0}, \boldsymbol{y} - \hat{\boldsymbol{y}})\right\| + \left\|\nabla\nabla_{\boldsymbol{y}} f\left(\boldsymbol{x}^{\theta}, \boldsymbol{y}\right)(\boldsymbol{0}, \boldsymbol{y} - \hat{\boldsymbol{y}})\right\| \\
&\quad + \frac{H_f}{2}\|\boldsymbol{y} - \hat{\boldsymbol{y}}\|^2 \\
&\le \left\|\nabla\nabla_{\boldsymbol{y}} f(\boldsymbol{x}_1, \bar{\boldsymbol{y}}_1) - \nabla\nabla_{\boldsymbol{y}} f\left(\boldsymbol{x}^{\theta}, \boldsymbol{y}\right)\right\| \|(\boldsymbol{0}, \boldsymbol{y} - \hat{\boldsymbol{y}})\| + \frac{H_f}{2}\|\boldsymbol{y} - \hat{\boldsymbol{y}}\|^2 + \frac{H_f}{2}\|\boldsymbol{y} - \hat{\boldsymbol{y}}\|^2 \\
&\le H_f\left(\|\boldsymbol{x}_1 - \boldsymbol{x}^{\theta}\| + \|\bar{\boldsymbol{y}}_1 - \boldsymbol{y}\|\right) \cdot \|\boldsymbol{y} - \hat{\boldsymbol{y}}\| + H_f\|\boldsymbol{y} - \hat{\boldsymbol{y}}\|^2 \\
&\le \frac{H_f}{2}\|\boldsymbol{x}_1 - \boldsymbol{x}^{\theta}\|^2 + \frac{H_f}{2}\|\bar{\boldsymbol{y}}_1 - \boldsymbol{y}\|^2 + H_f\|\boldsymbol{y} - \hat{\boldsymbol{y}}\|^2 + H_f\|\boldsymbol{y} - \hat{\boldsymbol{y}}\|^2 \\
&= \frac{H_f}{2}\|\boldsymbol{x}_1 - \boldsymbol{x}^{\theta}\|^2 + \frac{H_f}{2}\|\bar{\boldsymbol{y}}_1 - \boldsymbol{y}\|^2 + 2H_f\|\boldsymbol{y} - \hat{\boldsymbol{y}}\|^2,
\end{aligned}$$

where the second inequality uses the definition of matrix's $l_2$ norm and (22); the third inequality is due to the $H_f$-Lipschitz continuity of $\nabla\nabla_{\boldsymbol{y}} f$ and the triangle inequality; the forth inequality is due to the Cauchy inequality.

Combining the above estimate with $\|\boldsymbol{x}_1 - \boldsymbol{x}^{\theta}\| = (1-\theta)\|\boldsymbol{x}_1 - \boldsymbol{x}_2\|$, (20) (with $i = 1$), and the bound from (17), namely $\|\boldsymbol{y} - \hat{\boldsymbol{y}}\| \le 2(1-\theta)M_{\mathcal{S}}\|\boldsymbol{x}_1 - \boldsymbol{x}_2\|$, we obtain

$$\|\nabla_{\boldsymbol{y}} f(\boldsymbol{x}_1, \hat{\boldsymbol{y}}_1)\| \le \frac{H_f\left(1 + 17M_{\mathcal{S}}^2\right)}{2}(1-\theta)^2\|\boldsymbol{x}_1 - \boldsymbol{x}_2\|^2. \tag{23}$$

Finally, armed with the error bound condition in Assumption 1 (A3), we have

$$\text{dist}\left(\hat{\boldsymbol{y}}_1, \mathcal{S}(\boldsymbol{x}_1)\right) \le \frac{H_f\tau\left(1 + 17M_{\mathcal{S}}^2\right)}{2}(1-\theta)^2\|\boldsymbol{x}_1 - \boldsymbol{x}_2\|^2.$$

By the symmetric arguments, we can also have $\text{dist}\left(\hat{\boldsymbol{y}}_2, \mathcal{S}(\boldsymbol{x}_2)\right) \le H_f\tau\left(1 + 17M_{\mathcal{S}}^2\right)\theta^2\|\boldsymbol{x}_1 - \boldsymbol{x}_2\|^2/2$. We finished our proof for Claim 2. $\qquad\square$

## C.3 Step 3: Define approximation points

With the preparations in Steps 1–2, we now define the approximation points

$$\boldsymbol{y}_1 := \Pi_{\mathcal{S}(\boldsymbol{x}_1)}(\hat{\boldsymbol{y}}_1), \qquad \boldsymbol{y}_2 := \Pi_{\mathcal{S}(\boldsymbol{x}_2)}(\hat{\boldsymbol{y}}_2).$$

To establish the smoothness of $\mathcal{S}$, it remains to show that there exists a constant $L_{\mathcal{S}} > 0$ such that

$$\|\theta\boldsymbol{y}_1 + (1-\theta)\boldsymbol{y}_2 - \boldsymbol{y}\| \leq \frac{L_{\mathcal{S}}}{2}\theta(1-\theta)\|\boldsymbol{x}_1 - \boldsymbol{x}_2\|^2; \tag{24}$$

$$\|\boldsymbol{y}_1 - \boldsymbol{y}_2\|^2 \leq L_{\mathcal{S}}\|\boldsymbol{x}_1 - \boldsymbol{x}_2\|^2. \tag{25}$$

We first prove the inequality (24). We have

$$\begin{aligned}
&\|\theta\boldsymbol{y}_1 + (1-\theta)\boldsymbol{y}_2 - \boldsymbol{y}\| \\
=&\|\theta(\boldsymbol{y}_1 - \hat{\boldsymbol{y}}_1) + (1-\theta)(\boldsymbol{y}_2 - \hat{\boldsymbol{y}}_2) + (\theta\hat{\boldsymbol{y}}_1 + (1-\theta)\hat{\boldsymbol{y}}_2 - \boldsymbol{y})\| \\
\leq&\theta\|\boldsymbol{y}_1 - \hat{\boldsymbol{y}}_1\| + (1-\theta)\|\boldsymbol{y}_1 - \hat{\boldsymbol{y}}_1\| + \|(\theta\hat{\boldsymbol{y}}_1 + (1-\theta)\hat{\boldsymbol{y}}_2 - \boldsymbol{y})\| \\
=&\theta\operatorname{dist}(\hat{\boldsymbol{y}}_1, \mathcal{S}(\boldsymbol{x}_1)) + (1-\theta)\operatorname{dist}(\hat{\boldsymbol{y}}_2, \mathcal{S}(\boldsymbol{x}_2)) + \|\bar{\boldsymbol{y}}^\theta - \hat{\boldsymbol{y}}\| \\
=&\theta\operatorname{dist}(\hat{\boldsymbol{y}}_1, \mathcal{S}(\boldsymbol{x}_1)) + (1-\theta)\operatorname{dist}(\hat{\boldsymbol{y}}_2, \mathcal{S}(\boldsymbol{x}_2)) + \operatorname{dist}\left(\bar{\boldsymbol{y}}^\theta, \mathcal{S}\left(\boldsymbol{x}^\theta\right)\right),
\end{aligned}$$

where the second equality follows from $\boldsymbol{y}_i = \Pi_{\mathcal{S}(\boldsymbol{x}_i)}(\hat{\boldsymbol{y}}_i)$ and $\hat{\boldsymbol{y}}_i = \bar{\boldsymbol{y}}_i + \boldsymbol{y} - \hat{\boldsymbol{y}}$ for $i = 1, 2$; the final one is due to $\hat{\boldsymbol{y}} = \Pi_{\mathcal{S}(\boldsymbol{x}^\theta)}(\bar{\boldsymbol{y}}^\theta)$.

This, together with (15) and (18), implies

$$\|\theta\boldsymbol{y}_1 + (1-\theta)\boldsymbol{y}_2 - \boldsymbol{y}\| \leq (L_0 + L_1)\theta(1-\theta)\|\boldsymbol{x}_1 - \boldsymbol{x}_2\|^2.$$

Hence, (24) holds if $L_{\mathcal{S}} \geq 2(L_0 + L_1)$. It is left to show (25).

To prove (25), we estimate the distance $\|\boldsymbol{y}_1 - \boldsymbol{y}_2\|$. To begin, we apply the triangle inequality to obtain

$$\|\boldsymbol{y}_1 - \boldsymbol{y}_2\| \leq \|\boldsymbol{y}_1 - \bar{\boldsymbol{y}}_1\| + \|\bar{\boldsymbol{y}}_1 - \bar{\boldsymbol{y}}_2\| + \|\bar{\boldsymbol{y}}_2 - \boldsymbol{y}_2\|.$$

Then, use the definition $\boldsymbol{y}_i = \Pi_{\mathcal{S}(\boldsymbol{x}_i)}(\hat{\boldsymbol{y}}_i) = \Pi_{\mathcal{S}(\boldsymbol{x}_i)}(\bar{\boldsymbol{y}}_i + \boldsymbol{y} - \hat{\boldsymbol{y}})$ and notice $\bar{\boldsymbol{y}}_i \in \mathcal{S}(\boldsymbol{x}_i)$ for $i = 1, 2$. We have

$$\begin{aligned}
\|\boldsymbol{y}_1 - \boldsymbol{y}_2\| \leq & \left\|\Pi_{\mathcal{S}(\boldsymbol{x}_1)}(\bar{\boldsymbol{y}}_1 + (\boldsymbol{y} - \hat{\boldsymbol{y}})) - \Pi_{\mathcal{S}(\boldsymbol{x}_1)}(\bar{\boldsymbol{y}}_1))\right\| + \|\bar{\boldsymbol{y}}_1 - \bar{\boldsymbol{y}}_2\| \\
& + \left\|\Pi_{\mathcal{S}(\boldsymbol{x}_2)}(\bar{\boldsymbol{y}}_2)) - \Pi_{\mathcal{S}(\boldsymbol{x}_2)}(\bar{\boldsymbol{y}}_2 + (\boldsymbol{y} - \hat{\boldsymbol{y}}))\right\| \\
\leq & \|\hat{\boldsymbol{y}} - \boldsymbol{y}\| + \|\bar{\boldsymbol{y}}_1 - \bar{\boldsymbol{y}}_2\| + \|\hat{\boldsymbol{y}} - \boldsymbol{y}\| \\
= & 2\|\hat{\boldsymbol{y}} - \boldsymbol{y}\| + \|\bar{\boldsymbol{y}}_1 - \bar{\boldsymbol{y}}_2\|,
\end{aligned} \tag{26}$$

where the second inequality follows from the non-expansiveness of the projectors $\Pi_{\mathcal{S}(\boldsymbol{x}_1)}(\cdot)$ and $\Pi_{\mathcal{S}(\boldsymbol{x}_2)}(\cdot)$

Recall from (14) that $\|\bar{\boldsymbol{y}}_1 - \bar{\boldsymbol{y}}_2\| \leq M_{\mathcal{S}}\|\boldsymbol{x}_1 - \boldsymbol{x}_2\|$, and note that (17) further implies

$$\|\hat{\boldsymbol{y}} - \boldsymbol{y}\| \leq 2\theta(1-\theta)M_{\mathcal{S}}\|\boldsymbol{x}_1 - \boldsymbol{x}_2\| \leq \frac{1}{2}M_{\mathcal{S}}\|\boldsymbol{x}_1 - \boldsymbol{x}_2\|. \tag{27}$$

Combining (26) and (27) yields

$$\|\boldsymbol{y}_1 - \boldsymbol{y}_2\| \leq 2M_{\mathcal{S}}\|\boldsymbol{x}_1 - \boldsymbol{x}_2\|,$$

which proves (25) with $L_{\mathcal{S}} \geq 4M_{\mathcal{S}}^2$.

Finally, to ensure (24) it suffices to choose

$$L_{\mathcal{S}} \geq \max\{2(L_0 + L_1), 4M_{\mathcal{S}}^2\}.$$

Recalling $M_{\mathcal{S}} = L_f\tau$, $L_0 = \frac{H_f\tau}{2}(1 + M_{\mathcal{S}}^2)$, and $L_1 = \frac{H_f\tau}{2}(1 + 17M_{\mathcal{S}}^2)$, we obtain

$$L_{\mathcal{S}} = \max\left\{2H_f\tau\left(1 + 9L_f^2\tau^2\right), 4L_f^2\tau^2\right\}.$$

This completes the proof of Theorem 2.

### C.4 Proof of Remark 3

In the proof of Theorem 2, Assumption 1 is only used to guarantee Lemma 1 and the estimations on the involved points. Notice that to ensure Lemma 1, it suffices for Assumption 1 to hold on the set $\mathcal{D} \times \mathrm{Conv}(\bigcup_{\boldsymbol{x} \in \mathcal{D}} \mathcal{S}(\boldsymbol{x})) \subseteq \mathcal{D} \times \mathcal{Y}$. We only need to check that the involved points belong to the set $\mathcal{D} \times \mathcal{Y}$ to prove Remark 3.

As the domain of $\mathcal{S}$ is $\mathcal{D}$, to verify the set smoothness of $\mathcal{S}$, we can choose $\boldsymbol{x}_1, \boldsymbol{x}_2 \in \mathcal{D}$. Then, we have $\boldsymbol{x}^\theta \in \mathcal{D}$ due to the convexity of $\mathcal{D}$. Hence, it suffices to check that $\bar{\boldsymbol{y}}_1, \bar{\boldsymbol{y}}_2, \bar{\boldsymbol{y}}^\theta, \boldsymbol{y}_1, \boldsymbol{y}_2, \hat{\boldsymbol{y}}, \hat{\boldsymbol{y}}_1$, and $\hat{\boldsymbol{y}}_2$ belong to the convex set $\mathcal{Y}$.

First, recall that $\bar{\boldsymbol{y}}_1, \boldsymbol{y}_1 \in \mathcal{S}(\boldsymbol{x}_1), \bar{\boldsymbol{y}}_2, \boldsymbol{y}_2 \in \mathcal{S}(\boldsymbol{x}_2), \hat{\boldsymbol{y}} \in \mathcal{S}(\boldsymbol{x}^\theta)$, and $\bar{\boldsymbol{y}}^\theta \in \mathrm{Conv}(\mathcal{S}(\boldsymbol{x}_1) \cup \mathcal{S}(\boldsymbol{x}_2)) \subseteq \mathrm{Conv}(\bigcup_{\boldsymbol{x} \in \mathcal{D}} \mathcal{S}(\boldsymbol{x}))$ due to their definitions. We see that they belong to the set $\mathcal{Y}$. We then focus on $\hat{\boldsymbol{y}}_1$ and $\hat{\boldsymbol{y}}_2$. Recall that $\hat{\boldsymbol{y}}_i = \bar{\boldsymbol{y}}_i + (\boldsymbol{y} - \hat{\boldsymbol{y}})$ for $i = 1, 2$. We see that for $i = 1, 2$,

$$\mathrm{dist}\,(\hat{\boldsymbol{y}}_i, \mathcal{S}(\boldsymbol{x}_i)) \leq \|\hat{\boldsymbol{y}}_i - \bar{\boldsymbol{y}}_i\| = \|\boldsymbol{y} - \hat{\boldsymbol{y}}\|\,.$$

By (27) and $\|\boldsymbol{x}_1 - \boldsymbol{x}_2\| \leq \mathrm{diam}(\mathcal{D})$, for $i = 1, 2$, we further have

$$\mathrm{dist}\,(\hat{\boldsymbol{y}}_i, \mathcal{S}(\boldsymbol{x}_i)) \leq \frac{1}{2} M_{\mathcal{S}} \cdot \mathrm{diam}(\mathcal{D}).$$

This implies that $\hat{\boldsymbol{y}}_i \in \mathcal{S}(\boldsymbol{x}_i) + \frac{1}{2} M_{\mathcal{S}} \cdot \mathrm{diam}(\mathcal{D}) \cdot \mathbb{B}(\boldsymbol{0}, 1) \subseteq \mathcal{Y}$ for $i = 1, 2$. We complete the proof.

## D  Proof of Theorem 4

We first introduce some background on zeroth-order methods before the formal proof. Let $\mathbb{P}$ denote the uniform distribution on the unit sphere in $\mathbb{R}^m$. Given a function $g : \mathbb{R}^m \to \mathbb{R}$ and a radius $\varepsilon > 0$, we define the randomized smooth approximation $g$ by $g^\varepsilon(\boldsymbol{x}) := \mathbb{E}_{\boldsymbol{u} \sim \mathbb{P}}[g(\boldsymbol{x} + \varepsilon \boldsymbol{u})]$. We have the following properties for $g^\varepsilon$.

**Lemma 6** (Basic Properties of Randomized Smoothing)**.** *The following hold:*

(i) *If $g$ is $M_g$-Lipschitz continuous, then $g^\varepsilon$ is differentiable, $M_g$-Lipschitz continuous, and satisfies*

$$|g(\boldsymbol{x}) - g^\varepsilon(\boldsymbol{x})| \leq \varepsilon M_g. \tag{28}$$

(ii) *If $g$ is $r$-weakly convex (resp. concave), then $g^\varepsilon$ is $r$-weakly convex (resp. concave).*

*Proof.* (i) See [34, Proposition 2.3].

(ii) If $g$ is $r$-weakly convex, then by the same arguments of Nazari et al. [41, Lemma 16], we have the $r$-weak convexity of $g^\varepsilon$. When $g$ is $r$-weakly concave, we note that $-g$ is $r$-weakly convex, and hence $(-g)^\varepsilon$ is $r$-weakly convex. Due to the simple fact $(-g)^\varepsilon = -g^\varepsilon$, we have the $r$-weak convexity of $-g^\varepsilon$, i.e., the $r$-weak concavity of $g^\varepsilon$. $\qquad\square$

As the approximation for a subdifferential $\partial g$ via randomized smoothing can be inexact, we need the following $\epsilon$-subdifferential.

**Definition 4.** *Consider a convex function $g : \mathbb{R}^m \to \mathbb{R}$ and a scalar $\nu \geq 0$. We define the $\nu$-subdifferential of $g$ at $\boldsymbol{x} \in \mathbb{R}^m$ by*

$$\partial_\nu g(\boldsymbol{x}) = \left\{ \boldsymbol{s} \in \mathbb{R}^m : g(\boldsymbol{z}) \geq g(\boldsymbol{x}) + \boldsymbol{s}^T(\boldsymbol{z} - \boldsymbol{x}) - \nu, \ \forall\, \boldsymbol{z} \in \mathbb{R}^m \right\}.$$

We then develop the convergence rates of IZOM for optimistic and pessimistic BLO, respectively. We remark that our analysis remains valid when the hyper-objective functions $\varphi_o$ (resp. $\varphi_p$) are replaced with general $M$-Lipschitz continuous, $\rho$-weakly concave (resp. convex) functions.

### D.1  Optimistic Case

To begin, we record a celebrated proposition on subdifferential transportation.

**Proposition 1.** *(cf. [2, Theorem 5.5] and [43, Theorem 2]) Let $g : \mathbb{R}^m \to \mathbb{R}$ be a proper lower semicontinuous convex function. Suppose that $\nu \geq 0$ and $G \in \partial_\nu g(\boldsymbol{x})$. Then, for each $r > 0$, there is a unique vector $\boldsymbol{v} \in \mathbb{R}^m$ such that*

$$G - \frac{1}{r}\boldsymbol{v} \in \partial g(\boldsymbol{x} + r\boldsymbol{v}), \ \|\boldsymbol{v}\| \leq \sqrt{\nu}.$$

Proposition 1 plays an important role in relating an $\epsilon$-subdifferential to the Clarke subdifferential at a near point, leading to the following lemma.

**Lemma 7.** *Let $g : \mathbb{R}^m \to \mathbb{R}$ be an $M_g$-Lipschitz continuous and $\rho$-weakly concave function. Let $g^\varepsilon$ be the randomized approximation of $g$ with radius $\varepsilon > 0$ and $\nu = 2\varepsilon M_g$. Then, for all $\boldsymbol{x} \in \mathbb{R}^m$, we have*

$$\mathrm{dist}\left(\mathbf{0}, \bigcup_{\boldsymbol{z}\in\mathbb{B}(\boldsymbol{x},\sqrt{\nu})}\partial g(\boldsymbol{z})\right) \leq \|\nabla g^\varepsilon(\boldsymbol{x})\| + (\rho+1)\sqrt{\nu}. \tag{29}$$

*Proof of Lemma 7.* By Lemma 6 (ii), $g^\varepsilon$ is $\rho$-weakly concave, i.e., the function $\boldsymbol{x} \mapsto \frac{\rho}{2}\|\boldsymbol{x}\|^2 - g^\varepsilon(\boldsymbol{x})$ is convex. Then, we know that for all $\boldsymbol{z}, \boldsymbol{x} \in \mathbb{R}^m$,

$$\frac{\rho}{2}\|\boldsymbol{z}\|^2 - g^\varepsilon(\boldsymbol{z}) \geq \frac{\rho}{2}\|\boldsymbol{x}\|^2 - g^\varepsilon(\boldsymbol{x}) + (\rho\boldsymbol{x} - \nabla g^\varepsilon(\boldsymbol{x}))^T (\boldsymbol{z} - \boldsymbol{x}).$$

This, together with Lemma 6 (i), implies

$$\frac{\rho}{2}\|\boldsymbol{z}\|^2 - g(\boldsymbol{z}) \geq \frac{\rho}{2}\|\boldsymbol{x}\|^2 - g(\boldsymbol{x}) + (\rho\boldsymbol{x} - \nabla g^\varepsilon(\boldsymbol{x}))^T (\boldsymbol{z} - \boldsymbol{x}) - 2\varepsilon M_g,$$

which is equivalent to

$$\rho\boldsymbol{x} - \nabla g^\varepsilon(\boldsymbol{x}) \in \partial_\nu \left(\frac{\rho}{2}\|\boldsymbol{x}\|^2 - g(\boldsymbol{x})\right) \quad \text{with} \quad \nu = 2\varepsilon M_g.$$

For simplicity, we define $\bar{g} : \boldsymbol{x} \mapsto \frac{\rho}{2}\|\boldsymbol{x}\|^2 - g(\boldsymbol{x})$. Clearly, $\bar{g}$ is convex due to the $\rho$-weak concavity of $g$. Applying Proposition 1, we see that there exists $\boldsymbol{v} \in \mathbb{R}^m$ with $\|\boldsymbol{v}\| \leq \sqrt{\nu}$ such that

$$\rho\boldsymbol{x} - \nabla g^\varepsilon(\boldsymbol{x}) - \boldsymbol{v} \in \partial\bar{g}(\boldsymbol{x} + \boldsymbol{v}). \tag{30}$$

Recall that $-g$ is $\rho$-weakly convex, and thus is regular according to [49, Proposition 4.5]. Then, we have $\partial\bar{g}(\boldsymbol{z}) = \rho\boldsymbol{z} + \partial(-g)(\boldsymbol{z})$ for all $\boldsymbol{z} \in \mathbb{R}^m$ by [10, Corollary 3 of Proposition 2.3.3]. On the other hand, we have $\partial(-g) = -\partial g$ by [10, Proposition 2.3.1] and Lipschitz continuity of $g$. Hence, we see that

$$\partial\bar{g}(\boldsymbol{z}) = \rho\boldsymbol{z} - \partial g(\boldsymbol{z}), \qquad \forall\, \boldsymbol{z} \in \mathbb{R}^m.$$

In particular, we have

$$\partial\bar{g}(\boldsymbol{x} + \boldsymbol{v}) = \rho(\boldsymbol{x} + \boldsymbol{v}) - \partial g(\boldsymbol{x} + \boldsymbol{v}).$$

This, together with (30), implies

$$\nabla g^\varepsilon(\boldsymbol{x}) + (\rho+1)\boldsymbol{v} \in \partial g(\boldsymbol{x} + \boldsymbol{v}).$$

It follows that

$$\mathrm{dist}(\mathbf{0}, \partial g(\boldsymbol{x} + \boldsymbol{v})) \leq \|\nabla g^\varepsilon(\boldsymbol{x}) + (\rho+1)\boldsymbol{v}\| \leq \|\nabla g^\varepsilon(\boldsymbol{x})\| + (\rho+1)\|\boldsymbol{v}\|.$$

Observe that $\|\boldsymbol{v}\| \leq \sqrt{\nu}$ and

$$\mathrm{dist}\left(\mathbf{0}, \partial g(\boldsymbol{x} + \boldsymbol{v})\right) \geq \mathrm{dist}\left(\mathbf{0}, \bigcup_{\boldsymbol{z}\in\mathbb{B}(\boldsymbol{x},\|\boldsymbol{v}\|)}\partial g(\boldsymbol{z})\right) \geq \mathrm{dist}\left(\mathbf{0}, \bigcup_{\boldsymbol{z}\in\mathbb{B}(\boldsymbol{x},\sqrt{\nu})}\partial g(\boldsymbol{z})\right).$$

We obtain the desired inequality

$$\mathrm{dist}\left(\mathbf{0}, \bigcup_{\boldsymbol{z}\in\mathbb{B}(\boldsymbol{x},\sqrt{\nu})}\partial g(\boldsymbol{z})\right) \leq \|\nabla g^\varepsilon(\boldsymbol{x})\| + (\rho+1)\sqrt{\nu}.$$

$\square$

Now, we are ready to prove the convergence rate for IZOM. To begin, we define the subdifferential approximation function $G$ by

$$G(\boldsymbol{x}_t) = \frac{m}{2\varepsilon}(\varphi_o(\boldsymbol{x}_t + \varepsilon\boldsymbol{u}_t) - \varphi_o(\boldsymbol{x}_t - \varepsilon\boldsymbol{u}_t))\boldsymbol{u}_t.$$

By [34, Lemma D.1], it holds that

$$\mathbb{E}[G(\boldsymbol{x}_t)|\boldsymbol{x}_t] = \nabla\varphi_o^\varepsilon(\boldsymbol{x}_t); \quad \mathbb{E}[\|G(\boldsymbol{x}_t)\|^2|\boldsymbol{x}_t] \leq 16\sqrt{2\pi}mM_\varphi^2. \tag{31}$$

Since $|\tilde{\varphi}(\boldsymbol{x}_t) - \varphi_o(\boldsymbol{x}_t)| \leq w$ by the subroutine $\mathcal{A}$, we have $\|\tilde{G}(\boldsymbol{x}_t) - G(\boldsymbol{x}_t)\| \leq \frac{mw}{\varepsilon}$. This, together with the simple fact $\|\tilde{G}(\boldsymbol{x}_t)\|^2 \leq 2\|\tilde{G}(\boldsymbol{x}_t) - G(\boldsymbol{x}_t)\|^2 + 2\|G(\boldsymbol{x}_t)\|^2$ due to the Cauchy inequality, implies

$$\mathbb{E}[\|\tilde{G}(\boldsymbol{x}_t) - G(\boldsymbol{x}_t)\|^2|\boldsymbol{x}_t] \leq \left(\frac{mw}{\varepsilon}\right)^2; \quad \mathbb{E}[\|\tilde{G}(\boldsymbol{x}_t)\|^2|\boldsymbol{x}_t] \leq 32\sqrt{2\pi}mM_\varphi^2 + 2\left(\frac{mw}{\varepsilon}\right)^2. \tag{32}$$

Next, we combine the update of IZOM and $\rho$-weak concavity of $\varphi_o^\varepsilon$ to develop a sufficient decrease property for the $t$-th iteration. Using Fact 1 and the update $\boldsymbol{x}_{t+1} - \boldsymbol{x}_t = -\eta \tilde{G}(\boldsymbol{x}_t)$ of Algorithm 1, we obtain the following estimate:

$$\varphi_o^\varepsilon(\boldsymbol{x}_{t+1})$$
$$\leq \varphi_o^\varepsilon(\boldsymbol{x}_t) + \nabla\varphi_o^\varepsilon(\boldsymbol{x}_t)^T(\boldsymbol{x}_{t+1} - \boldsymbol{x}_t) + \frac{\rho}{2}\|\boldsymbol{x}_{t+1} - \boldsymbol{x}_t\|^2$$
$$= \varphi_o^\varepsilon(\boldsymbol{x}_t) - \eta\nabla\varphi_o^\varepsilon(\boldsymbol{x}_t)^T\tilde{G}(\boldsymbol{x}_t) + \frac{\rho}{2}\eta^2\|\tilde{G}(\boldsymbol{x}_t)\|^2$$
$$= \varphi_o^\varepsilon(\boldsymbol{x}_t) - \eta\nabla\varphi_o^\varepsilon(\boldsymbol{x}_t)^T G(\boldsymbol{x}_t) - \eta\nabla\varphi_o^\varepsilon(\boldsymbol{x}_t)^T\left(\tilde{G}(\boldsymbol{x}_t) - G(\boldsymbol{x}_t)\right) + \frac{\rho}{2}\eta^2\|\tilde{G}(\boldsymbol{x}_t)\|^2$$
$$\leq \varphi_o^\varepsilon(\boldsymbol{x}_t) - \eta\nabla\varphi_o^\varepsilon(\boldsymbol{x}_t)^T G(\boldsymbol{x}_t) + \frac{\eta}{2}\|\nabla\varphi_o^\varepsilon(\boldsymbol{x}_t)\|^2 + \frac{\eta}{2}\left\|\tilde{G}(\boldsymbol{x}_t) - G(\boldsymbol{x}_t)\right\|^2 + \frac{\rho}{2}\eta^2\|\tilde{G}(\boldsymbol{x}_t)\|^2.$$

where the last inequality is due to the Cauchy inequality.

We take expectation conditioning on $\boldsymbol{x}_t$ for the above inequality. Recall that $\mathbb{E}[G(\boldsymbol{x}_t)|\boldsymbol{x}_t] = \nabla\varphi_o^\varepsilon(\boldsymbol{x}_t)$ by (31). We see that

$$\mathbb{E}[\varphi_o^\varepsilon(\boldsymbol{x}_{t+1})|\boldsymbol{x}_t] \leq \varphi_o^\varepsilon(\boldsymbol{x}_t) - \frac{\eta}{2}\|\nabla\varphi_o^\varepsilon(\boldsymbol{x}_t)\|^2 + \frac{\eta}{2}\mathbb{E}[\|\tilde{G}(\boldsymbol{x}_t) - G(\boldsymbol{x}_t)\|^2|\boldsymbol{x}_t]$$
$$+ \frac{\rho\eta^2}{2}\mathbb{E}[\|\tilde{G}(\boldsymbol{x}_t)\|^2|\boldsymbol{x}_t].$$

Apply Lemma 7 to $\varphi_o$ and use the Cauchy inequality. We obtain

$$\mathrm{dist}\left(\boldsymbol{0}, \bigcup_{\boldsymbol{z}\in\mathbb{B}(\boldsymbol{x}_t, \sqrt{\nu})}\partial\varphi_o(\boldsymbol{z})\right)^2 \leq 2\|\nabla\varphi_o^\varepsilon(\boldsymbol{x}_t)\|^2 + 2(\rho+1)^2\varepsilon.$$

Combining the above two inequalities with (32), we have

$$\mathbb{E}[\varphi_o^\varepsilon(\boldsymbol{x}_{t+1})|\boldsymbol{x}_t] \leq \varphi_o^\varepsilon(\boldsymbol{x}_t) - \frac{\eta}{4}\mathrm{dist}\left(\boldsymbol{0}, \bigcup_{\boldsymbol{z}\in\mathbb{B}(\boldsymbol{x}_t, \sqrt{\nu})}\partial\varphi_o(\boldsymbol{z})\right)^2 + \frac{\eta}{2}(\rho+1)^2\varepsilon$$
$$+ \frac{\eta}{2}\left(\frac{mw}{\varepsilon}\right)^2 + \frac{\rho}{2}\eta^2\left(32\sqrt{2\pi}mM_\varphi^2 + 2\left(\frac{mw}{\varepsilon}\right)^2\right).$$

Recall that $\nu = 2\varepsilon M_\varphi$, $\eta = \Theta(\frac{1}{\sqrt{mT}})$, $\varepsilon = \mathcal{O}(\frac{1}{\sqrt{T}})$, and $w = \mathcal{O}(\frac{1}{m^{\frac{3}{4}}T^{\frac{3}{4}}})$. Ignoring some scalars, we further have

$$\mathbb{E}[\varphi_o^\varepsilon(\boldsymbol{x}_{t+1})|\boldsymbol{x}_t] \leq \varphi_o^\varepsilon(\boldsymbol{x}_t) - \frac{\eta}{4}\mathrm{dist}\left(\boldsymbol{0}, \bigcup_{\boldsymbol{z}\in\mathbb{B}(\boldsymbol{x}_t, \sqrt{\nu})}\partial\varphi_o(\boldsymbol{z})\right)^2 + \mathcal{O}\left(\frac{1}{T}\right). \quad (33)$$

Summing (33) over $t = 0, 1, \ldots T-1$ and taking full expectation, we obtain

$$\mathbb{E}[\varphi_o^\varepsilon(\boldsymbol{x}_T)] \leq \varphi_o^\varepsilon(\boldsymbol{x}_0) - \frac{\eta}{4}\sum_{t=0}^{T-1}\mathbb{E}\left[\mathrm{dist}\left(\boldsymbol{0}, \bigcup_{\boldsymbol{z}\in\mathbb{B}(\boldsymbol{x}_t, \sqrt{\nu})}\partial\varphi_o(\boldsymbol{z})\right)^2\right] + \mathcal{O}(1).$$

Note that the definition of $\bar{\boldsymbol{x}}$ yields

$$\mathbb{E}\left[\mathrm{dist}\left(\boldsymbol{0}, \bigcup_{\boldsymbol{z}\in\mathbb{B}(\bar{\boldsymbol{x}}, \sqrt{\nu})}\partial\varphi_o(\boldsymbol{z})\right)^2\right] = \frac{1}{T}\sum_{t=0}^{T-1}\mathbb{E}\left[\mathrm{dist}\left(\boldsymbol{0}, \bigcup_{\boldsymbol{z}\in\mathbb{B}(\boldsymbol{x}_t, \sqrt{\nu})}\partial\varphi_o(\boldsymbol{z})\right)^2\right].$$

On the other hand, by Lemma 6 (i) and $M_\varphi$-Lipschitz continuity of $\varphi_o$,

$$\varphi_o^\varepsilon(\boldsymbol{x}_0) - \mathbb{E}[\varphi_o^\varepsilon(\boldsymbol{x}_T)] \leq 2\varepsilon M_\varphi + \varphi_o(\boldsymbol{x}_0) - \min_{\boldsymbol{x}}\varphi_o(\boldsymbol{x}) = \Delta_o.$$

Putting all the things together, we have

$$\mathbb{E}\left[\mathrm{dist}\left(\boldsymbol{0}, \bigcup_{\boldsymbol{z}\in\mathbb{B}(\bar{\boldsymbol{x}}, \sqrt{\nu})}\partial\varphi_o(\boldsymbol{z})\right)^2\right] = \mathcal{O}\left(\frac{\Delta_o + 1}{\eta T}\right) = \mathcal{O}\left(\frac{\sqrt{m}(\Delta_o + 1)}{\sqrt{T}}\right),$$

where $\sqrt{\nu} = \sqrt{2\varepsilon M_\varphi} = \mathcal{O}(T^{-\frac{1}{4}})$.

## D.2 Pessimistic Case

In this section, we let $G(\boldsymbol{x}_t) = \frac{m}{2\varepsilon}(\varphi_p(\boldsymbol{x}_t + \varepsilon\boldsymbol{u}_t) - \varphi_p(\boldsymbol{x}_t - \varepsilon\boldsymbol{u}_t))\boldsymbol{u}_t$ and use $\hat{\boldsymbol{x}}$ to denote $\mathrm{prox}_{\gamma,\varphi_p}(\boldsymbol{x})$ for simplicity. Similar to the arguments on (31), (32), we have the following due to [34, Lemma D.1] and $|\tilde{\varphi}(\boldsymbol{x}_t) - \varphi_p(\boldsymbol{x}_t)| \le w$ given by Algorithm 1:

$$\mathbb{E}[G(\boldsymbol{x}_t)|\boldsymbol{x}_t] = \nabla\varphi_p^\varepsilon(\boldsymbol{x}_t); \quad \mathbb{E}[\|G(\boldsymbol{x}_t)\|^2|\boldsymbol{x}_t] \le 16\sqrt{2\pi}mM_\varphi^2. \tag{34}$$

$$\mathbb{E}[\|\tilde{G}(\boldsymbol{x}_t) - G(\boldsymbol{x}_t)\|^2|\boldsymbol{x}_t] \le \left(\frac{mw}{\varepsilon}\right)^2; \quad \mathbb{E}[\|\tilde{G}(\boldsymbol{x}_t)\|^2|\boldsymbol{x}_t] \le 32\sqrt{2\pi}mM_\varphi^2 + 2\left(\frac{mw}{\varepsilon}\right)^2. \tag{35}$$

Invoking the methodology of Davis and Drusvyatskiy [12], we first estimate $\|\hat{\boldsymbol{x}}_t - \boldsymbol{x}_{t+1}\|$ for $t = 0, 1, \ldots, T-1$. To begin, the update of Algorithm 1 and direct computation give the following estimate:

$$\|\hat{\boldsymbol{x}}_t - \boldsymbol{x}_{t+1}\|^2$$
$$= \|\hat{\boldsymbol{x}}_t - \boldsymbol{x}_t + \eta\tilde{G}(\boldsymbol{x}_t)\|$$
$$= \|\hat{\boldsymbol{x}}_t - \boldsymbol{x}_t\|^2 + \eta^2\|\tilde{G}(\boldsymbol{x}_t)\|^2 + 2\eta\tilde{G}(\boldsymbol{x}_t)^T(\hat{\boldsymbol{x}}_t - \boldsymbol{x}_t)$$
$$= \|\hat{\boldsymbol{x}}_t - \boldsymbol{x}_t\|^2 + \eta^2\|\tilde{G}(\boldsymbol{x}_t)\|^2 + 2\eta G(\boldsymbol{x}_t)^T(\hat{\boldsymbol{x}}_t - \boldsymbol{x}_t) + 2\eta\left(\tilde{G}(\boldsymbol{x}_t) - G(\boldsymbol{x}_t)\right)^T(\hat{\boldsymbol{x}}_t - \boldsymbol{x}_t)$$
$$\le \|\hat{\boldsymbol{x}}_t - \boldsymbol{x}_t\|^2 + \eta^2\|\tilde{G}(\boldsymbol{x}_t)\|^2 + 2\eta G(\boldsymbol{x}_t)^T(\hat{\boldsymbol{x}}_t - \boldsymbol{x}_t) + \eta\|\tilde{G}(\boldsymbol{x}_t) - G(\boldsymbol{x}_t)\|^2 + \eta\|\hat{\boldsymbol{x}}_t - \boldsymbol{x}_t\|^2. \tag{36}$$

Taking expectation in (36) and using (34), (35), we obtain the following inequality:

$$\mathbb{E}[\|\hat{\boldsymbol{x}}_t - \boldsymbol{x}_{t+1}\|^2|\boldsymbol{x}_t]$$
$$\le (1+\eta)\|\hat{\boldsymbol{x}}_t - \boldsymbol{x}_t\|^2 + 2\eta\nabla\varphi_p^\varepsilon(\boldsymbol{x}_t)^T(\hat{\boldsymbol{x}}_t - \boldsymbol{x}_t) + \eta\,\mathbb{E}[\|\tilde{G}(\boldsymbol{x}_t) - G(\boldsymbol{x}_t)\|^2|\boldsymbol{x}_t]$$
$$\quad + \eta^2\,\mathbb{E}[\|\tilde{G}(\boldsymbol{x}_t)\|^2|\boldsymbol{x}_t]$$
$$\le (1+\eta)\|\hat{\boldsymbol{x}}_t - \boldsymbol{x}_t\|^2 + 2\eta\nabla\varphi_p^\varepsilon(\boldsymbol{x}_t)^T(\hat{\boldsymbol{x}}_t - \boldsymbol{x}_t) + \eta\left(\frac{mw}{\varepsilon}\right)^2 + 32\sqrt{2\pi}mM_\varphi^2\eta^2 + 2\left(\frac{mw}{\varepsilon}\right)^2\eta^2. \tag{37}$$

We then turn to estimate $\varphi_{p,\gamma}(\boldsymbol{x}_{t+1})$. By the definition of $\varphi_{p,\gamma}$, we have

$$\varphi_{p,\gamma}(\boldsymbol{x}_{t+1}) \le \varphi_p(\hat{\boldsymbol{x}}_t) + \frac{1}{2\gamma}\|\hat{\boldsymbol{x}}_t - \boldsymbol{x}_{t+1}\|^2.$$

This, together with (37), implies

$$\mathbb{E}[\varphi_{p,\gamma}(\boldsymbol{x}_{t+1})|\boldsymbol{x}_t] - \varphi_p(\hat{\boldsymbol{x}}_t)$$
$$\le \frac{1+\eta}{2\gamma}\|\hat{\boldsymbol{x}}_t - \boldsymbol{x}_t\|^2 + \frac{\eta}{\gamma}\nabla\varphi_p^\varepsilon(\boldsymbol{x}_t)^T(\hat{\boldsymbol{x}}_t - \boldsymbol{x}_t) + \frac{\eta}{2\gamma}\left(\frac{mw}{\varepsilon}\right)^2 + 16\sqrt{2\pi}mM_\varphi^2\frac{\eta^2}{\gamma} + \left(\frac{mw}{\varepsilon}\right)^2\frac{\eta^2}{\gamma}.$$

Recall that $\varphi_p^\varepsilon$ is $\rho$-weakly convex by Theorem 3 and Lemma 6 (ii). Then, using Lemma 3 (iii), we have

$$\nabla\varphi_p^\varepsilon(\boldsymbol{x}_t)^T(\hat{\boldsymbol{x}}_t - \boldsymbol{x}_t) \le \varphi_p^\varepsilon(\hat{\boldsymbol{x}}_t) - \varphi_p^\varepsilon(\boldsymbol{x}_t) + \frac{\rho}{2}\|\hat{\boldsymbol{x}}_t - \boldsymbol{x}_t\|^2$$
$$\le \varphi_p(\hat{\boldsymbol{x}}_t) - \varphi_p(\boldsymbol{x}_t) + \frac{\rho}{2}\|\hat{\boldsymbol{x}}_t - \boldsymbol{x}_t\|^2 + 2\varepsilon M_\varphi,$$

where the second inequality is due to Lemma 6 (i).

Combining the above two estimates gives

$$\mathbb{E}[\varphi_{p,\gamma}(\boldsymbol{x}_{t+1})|\boldsymbol{x}_t]$$
$$\le \varphi_p(\hat{\boldsymbol{x}}_t) + \frac{1}{2\gamma}\|\hat{\boldsymbol{x}}_t - \boldsymbol{x}_t\|^2 + \frac{\eta}{\gamma}\left(\varphi_p(\hat{\boldsymbol{x}}_t) - \varphi_p(\boldsymbol{x}_t) + \frac{\rho+1}{2}\|\hat{\boldsymbol{x}}_t - \boldsymbol{x}_t\|^2\right) + \frac{2\eta}{\gamma}\varepsilon M_\varphi$$
$$\quad + \frac{\eta}{2\gamma}\left(\frac{mw}{\varepsilon}\right)^2 + 16\sqrt{2\pi}mM_\varphi^2\frac{\eta^2}{\gamma} + \left(\frac{mw}{\varepsilon}\right)^2\frac{\eta^2}{\gamma}$$
$$= \varphi_{p,\gamma}(\boldsymbol{x}_t) + \frac{\eta}{\gamma}\left(\varphi_{p,\gamma}(\boldsymbol{x}_t) - \varphi_p(\boldsymbol{x}_t) + \frac{(\rho+1)\gamma-1}{2\gamma}\|\hat{\boldsymbol{x}}_t - \boldsymbol{x}_t\|^2\right) + \frac{2\eta}{\gamma}\varepsilon M_\varphi$$
$$\quad + \frac{\eta}{2\gamma}\left(\frac{mw}{\varepsilon}\right)^2 + 16\sqrt{2\pi}mM_\varphi^2\frac{\eta^2}{\gamma} + \left(\frac{mw}{\varepsilon}\right)^2\frac{\eta^2}{\gamma},$$

where the equation uses $\varphi_{p,\gamma}(\boldsymbol{x}_t) = \varphi_p(\hat{\boldsymbol{x}}_t) + \frac{1}{2\gamma}\|\hat{\boldsymbol{x}}_t - \boldsymbol{x}_t\|^2$. Recall that by Lemma 4, it holds that $\boldsymbol{x}_t - \hat{\boldsymbol{x}}_t = \gamma\nabla\varphi_{p,\gamma}(\boldsymbol{x}_t)$ and

$$\varphi_{p,\gamma}(\boldsymbol{x}_t) - \varphi_p(\boldsymbol{x}_t) \leq -\frac{1-\gamma\rho}{2\gamma}\|\hat{\boldsymbol{x}}_t - \boldsymbol{x}_t\|^2 \leq -\frac{1-\gamma(\rho+1)}{2\gamma}\|\hat{\boldsymbol{x}}_t - \boldsymbol{x}_t\|^2.$$

We further have

$$\mathbb{E}[\varphi_{p,\gamma}(\boldsymbol{x}_{t+1})|\boldsymbol{x}_t] \leq \varphi_{p,\gamma}(\boldsymbol{x}_t) - \eta\left(1 - \gamma(\rho+1)\right)\|\nabla\varphi_{p,\gamma}(\boldsymbol{x}_t)\|^2 + \frac{2\eta}{\gamma}\varepsilon M_\varphi$$
$$+ \frac{\eta}{2\gamma}\left(\frac{mw}{\varepsilon}\right)^2 + 16\sqrt{2\pi}mM_\varphi^2\frac{\eta^2}{\gamma} + \left(\frac{mw}{\varepsilon}\right)^2\frac{\eta^2}{\gamma}.$$

Recall that $\eta = \Theta(\frac{1}{\sqrt{mT}}), \varepsilon = \mathcal{O}(\frac{1}{\sqrt{T}}), w = \mathcal{O}(\frac{1}{m^{\frac{3}{4}}T^{\frac{3}{4}}})$, and $\gamma \in (0, 1/(\rho+1))$. Neglecting some scalars, it follows that

$$\mathbb{E}[\varphi_{p,\gamma}(\boldsymbol{x}_{t+1})|\boldsymbol{x}_t] \leq \varphi_{p,\gamma}(\boldsymbol{x}_t) - \eta\left(1 - \gamma(\rho+1)\right)\|\nabla\varphi_{p,\gamma}(\boldsymbol{x}_t)\|^2 + \mathcal{O}\left(\frac{1}{T}\right). \tag{38}$$

Summing (38) over $t = 0, 1, \ldots T - 1$ and taking full expectation, we have

$$\mathbb{E}[\varphi_{p,\gamma}(\boldsymbol{x}_T)] \leq \varphi_{p,\gamma}(\boldsymbol{x}_0) - \eta\left(1 - \gamma(\rho+1)\right)\sum_{t=0}^{T-1}\mathbb{E}[\|\nabla\varphi_{p,\gamma}(\boldsymbol{x}_t)\|^2] + \mathcal{O}(1).$$

Note that the definition of $\bar{\boldsymbol{x}}$ gives $\sum_{t=0}^{T-1}\|\nabla\varphi_{p,\gamma}(\boldsymbol{x}_t)\|^2 = T\,\mathbb{E}[\|\nabla\varphi_{p,\gamma}(\bar{\boldsymbol{x}})\|^2]$. Also, notice that

$$\varphi_{p,\gamma}(\boldsymbol{x}_0) - \mathbb{E}[\varphi_{p,\gamma}(\boldsymbol{x}_T)] \leq \varphi_{p,\gamma}(\boldsymbol{x}_0) - \min_{\boldsymbol{x}}\varphi_{p,\gamma}(\boldsymbol{x}) = \Delta_p.$$

Putting all the things together, we obtain

$$\mathbb{E}[\|\nabla\varphi_{p,\gamma}(\bar{\boldsymbol{x}})\|^2] = \mathcal{O}\left(\frac{\Delta_p + 1}{\eta T}\right) = \mathcal{O}\left(\frac{\sqrt{m}(\Delta_p + 1)}{\sqrt{T}}\right).$$

This completes the proof.

