# OpenReview forum: "Set Smoothness Unlocks Clarke Hyper-stationarity in Bilevel Optimization"
_NeurIPS.cc/2025/Conference — NeurIPS 2025 spotlight_

### Official Review · Reviewer_FxiL · 2025-06-10

**Clarity:** 3
**Significance:** 4
**Originality:** 3
**Rating:** 5
**Confidence:** 3

**Summary:**

This paper studies bilevel optimization. The authors introduce a novel notion of set smoothness to characterize the solution set and hyper objective. Using this set smoothness, the authors prove that under a PL-like assumption, the hyper objective is weak convex(concave), enabling the computing of a Clarke hyper stationary point. The authors propose a zero-th order algorithm to compute it non-asymptotically.

**Questions:**

Do you think the zero-th order algorithm have implications for first order algorithms, which is more practical? What is the major gap between them?

**Ethical Concerns:**

["NO or VERY MINOR ethics concerns only"]

**Final Justification:**

I will keep my score.

**Limitations:**

yes

**Paper Formatting Concerns:**

This paper do not use the provided format.

**Quality:**

4

**Strengths And Weaknesses:**

The main strength of this paper is the theoretical contribution. The establishment of weak convexity (concavity) and computability of Clarke hyper stationary under PL-like condition is a novel and worthwhile contribution. The introduction of the set smoothness has technical novelty and implications for future work.

The weakness of the paper lies in its proposed algorithm. It is a zero-th algorithm and may not be practically efficient. The authors also do not provide empirical results.

---

> ### Author Rebuttal · Authors · 2025-07-28
>
> We express our sincere gratitude to reviewer FxiL for recognizing our contributions and insightful comments. Below, we address the concerns raised.
>
> **On the practical efficiency of the zero-order method**
>
> In the algorithmic part of our work, the primary goal is establishing the theoretical computability of Clarke hyper-stationarity for the concerned BLO. To this end, it suffices to adopt a standard algorithm, i.e., a zero-order method, and utilize the developed structural properties. We view this computability result as a foundational step toward designing more efficient algorithms for computing Clarke hyper-stationary points. We did not provide empirical results in the current version because, to the best of our knowledge, no benchmark algorithms yet exist for computing Clarke hyper-stationary points. Given more space in the main body, we may include emprical experiments in a future version of our manuscript.
>
> |*Do you think the zero-th order algorithm have implications for first order algorithms, which is more practical? What is the major gap between them?*
>
> Certainly. We believe that the convergence guarantees established for the zero-order algorithm offer meaningful implications for first-order methods. In particular, one may expect stronger results—such as better dependence on the parameters and dimension—when using first-order algorithms.
>
> Despite the potential efficiency of first-order methods, the evaluation of the hyper-subgradient might be a challenge to implement them, as this involves computing the subgradient of parametric function $\min_{y\in S(x)} F(x,y)$ or $\max_{y\in S(x)} F(x,y)$. This is highly nontrivial especially when $S(x)$ has multiple elements. In comparison,zero-order methods only require evaluations of the hyper-objective value, which are readily approximated using simple bilevel algorithms, as we discussed in lines 297-298 and Footnote 3.
>
> That said, exploring first-order methods for computing Clarke hyper-stationarity in BLO with multiple lower-level solutions is a promising direction. In particular, once an efficient oracle for computing the hyper-subgradient was developed, it may lead to stronger convergence guarantees and practically effective algorithms for computing Clarke hyper-stationarity.
>
> We thank the reviewer for pointing out the format issue. We used the provided neurips_2025.sty file in our tex file without modifying the margins or font size. We will address any remaining minor discrepancies in a future version of our manuscript.

---

> > ### Comment · Reviewer_FxiL · 2025-08-05
> >
> > Thanks for the response, I will keep my score.

---

### Official Review · Reviewer_PTCZ · 2025-06-16

**Clarity:** 3
**Significance:** 3
**Originality:** 3
**Rating:** 5
**Confidence:** 3

**Summary:**

This paper proves that strong Clarke hyper-stationary points are computable for nonconvex-PL bilevel problems. The main results build on the following three arguments:
1. The solution set of a smooth PL function satisfies the set smoothness introduced in this paper (Proposition 2).
2. The hyper-objective is weakly convex/concave when the lower-level solution set is smooth (Proposition 1).
3. A zeroth-order method can find strong Clarke stationary points for a weakly convex/concave objective (Section 4).

All the above arguments are novel. I especially appreciate Proposition 1 and Proposition 2, which are highly non-trivial and I think they definitely contribute a deeper understanding of bilevel optimization and other fields of optimization theory.

**Questions:**

I do not have additional questions.

**Ethical Concerns:**

["NO or VERY MINOR ethics concerns only"]

**Limitations:**

One limitation is the dimension dependency in the upper bound, which comes from the use of the zeroth-order method. I can understand that the dimension dependency is hard to remove, but it is still a limitation compared to the existing dimension-free upper bounds (although they may require additional assumptions).

**Paper Formatting Concerns:**

I do not notice any major formatting issues.

**Quality:**

3

**Strengths And Weaknesses:**

The strengths of this paper are clear:
1. The paper is well written.
2. Proposition 1 and Proposition 2 are very novel and both solve important theoretical problems.

I do not notice significant weaknesses; the only one I find is contained in the limitation part.

---

> ### Author Rebuttal · Authors · 2025-07-28
>
> We sincerely thank reviewer PTCZ for appreciating our contributions to bilevel optimization and providing valuable feedback. We also hope that Proposition 1 and Proposition 2 will find their applications in other optimization fields. Below, we address the concern raised.
>
> **On the dimension dependency in the upper bound**
>
> Thank you for the insightful observation and valuable comments. In Sec. 4, our primary goal is to establish the computability of Clarke stationarity, rather than to optimize the dimension dependence in the upper bound. That said, we agree that improving the dimension dependence is a compelling direction for future work. As you suggested, it may be possible to obtain a dimension-free upper bound under additional structural assumptions, which we believe is worth exploring further.

---

### Official Review · Reviewer_orA8 · 2025-07-01

**Clarity:** 4
**Significance:** 3
**Originality:** 4
**Rating:** 5
**Confidence:** 3

**Summary:**

The paper considers bilevel optimization problems in the non-convex/PL setting where the lower-level solution set is not necessarily a singleton. The authors introduce the notion of set smoothness and show that the solution set of the inner function in the considered setting is smooth. Then, they show that this smoothness implies the weak convexity (resp. the weak concavity) of the pessimistic (resp. optimistic) hyper-objective. With this regularity, they show that Clark stationarity can be obtained with a zeroth-order algorithm.

**Questions:**

N/A

**Ethical Concerns:**

["NO or VERY MINOR ethics concerns only"]

**Final Justification:**

The paper introduces the new concept of set smoothness, which opens the way for new analyses of bilevel optimization problems when the inner function is non-convex. For the novelty and the potential significance of this paper, I recommend acceptance.

**Quality:**

4

**Strengths And Weaknesses:**

### Strengths

- **S1**: The paper is very well-written.

- **S2**: The paper introduce the concept of set smoothness which offers new perspectives in analyzing bilevel optimization when the inner function is not strongly convex.

- **S3**: The authors establish regularity properties of the hyper-objective based on the introduced set smoothness.

- **S4**: The authors prove computability of Clark stationarity for non-smooth hyperobjectives.


### Weaknesses

- **W1**: The assumption on the convexity of $\mathcal{S}(x)$ is strong for a non-convex inner function.



#### Minor/Typo
- For clarity, it would be useful to recall the definition of Goldstein stationarity
- **Line 186**: "Lemma 2 (ii) and (iv)" -> "Lemma 3 (ii) and (iv)"
- **Line 496**: "$\varphi(x_2)$" -> "$\varphi_p(x_2)$"

---

> ### Author Rebuttal · Authors · 2025-07-28
>
> We sincerely thank reviewer orA8 for recognizing our contributions and providing insightful comments. Below, we address the concerns raised.
>
> **On the convexity assumption on $S(x)$**
>
> Thanks for pointing out this. We believe that this assumption is natural in BLO setting because defining the hyper-objective function involves minimizing the upper-level function over $S(x)$. If $S(x)$ were nonconvex, even evaluating the hyper-objective value generally become NP-hard. Furthermore, this assumption is weaker than the convexity of the lower-level function adopted in the literature (see e.g., [7, Assumption E.1], [42, Assumption 3.4], and [16, Chapter 9.1]).
>
> Nevertheless, one can still weaken this convexity assumption. In our analysis, this convexity condition is primarily used to ensure the non-expansive property of the projection operator $\Pi_{S(x)}$ in lines 580 and 620. Thus, it be replaced by a weaker condition, such as a non-expansive-type property of the projection, without affecting our main results in Sec. 3.
>
> **On the minor issues and typo**
>
> We appreciate the reviewer's careful reading and pointing out the two typos. We will correct them accordingly. Also, given more space for the main body, we will include a brief recall of the definition of Goldstein stationarity for clarity in a future version of the manuscript.

---

> > ### Comment · Reviewer_orA8 · 2025-08-04
> >
> > I thank the authors for their answer. I keep my positive score.

---

### Official Review · Reviewer_fL5D · 2025-07-02

**Clarity:** 3
**Significance:** 3
**Originality:** 4
**Rating:** 5
**Confidence:** 4

**Summary:**

This paper investigates unconstrained bilevel optimization problems in which the lower-level problem admits multiple solutions. The central contribution is the introduction of a novel set-smoothness condition on the lower-level solution mapping. The authors show that under this condition, the associated hyper-objectives -- whether defined optimistically or pessimistically -- exhibit weak concavity (or weak convexity in the pessimistic case). This structural property enables the use of subgradient-based methods to find approximate Clarke stationary points efficiently. To this end, the authors develop a zeroth-order algorithm tailored for the bilevel setting.

**Questions:**

- Has Definition 2 (set-smoothness) appeared in prior literature, perhaps in the context of set-valued analysis or variational analysis? Understanding any precedent could help position the contribution more broadly.

- Could the proposed method or analysis be extended to stochastic Bilevel settings?

**Ethical Concerns:**

["NO or VERY MINOR ethics concerns only"]

**Final Justification:**

I believe the potential of their conceptual contributions.

**Limitations:**

This is a well-written and insightful paper that introduces a novel structural condition with meaningful implications for Bilevel optimization. While the algorithmic contributions are incremental, the conceptual clarity and technical rigor make it a valuable addition to the literature. I recommend acceptance.

**Paper Formatting Concerns:**

The used font seems different from the default setting.

**Quality:**

3

**Strengths And Weaknesses:**

**Strengths**

To the best of my knowledge, this is the first paper where the notion of set-smoothness has been introduced in the context of Bilevel optimization. The authors effectively leverage this idea to unify and strengthen several threads in nonsmooth optimization -- namely, Clarke stationarity, weak convexity, and zeroth-order methods -- to tackle a longstanding challenge in bilevel problems with multiple lower-level solutions. The presentation is concise and the logical development of the core ideas is clear and well-structured.


**Weaknesses**

The main conceptual contribution is compelling, but the following must be clarified:

- The core contribution of this submission lies in the introduction of the set-smoothness property, and I believe it might have significant future potential to advance understanding in multiple-solution settings — an area where progress has been notably limited. That said, to fully realize this potential, the submission should more clearly position the proposed condition within the broader context and articulate its implications in greater depth. In particular, it would be essential to discuss the algorithmic implication of results in Section 3 in greater details.


Also, the algorithmic developments could be more thoroughly distinguished and emphasized:

- While the paper builds toward Clarke stationarity, the motivation for preferring Clarke subgradients over alternatives such as the Goldstein subdifferential could be more fully justified. Is this choice driven purely by analytical tractability, or does it yield practical benefits (e.g., stronger convergence guarantees, better empirical performance)?

- The distinction between the proposed Algorithm 1 and the approach in [7] could be made more transparent. A line-by-line comparison or at least a clear summary of the differences would be helpful. If empirical performance comparisons are available, they would strengthen the case.

- A discussion of how general the set-smoothness condition is would be valuable. For example, how does it relate to more common regularity conditions like error bounds or the Polyak-Łojasiewicz condition at the lower level? Are there natural examples where set-smoothness holds but other assumptions fail?

---

> ### Author Rebuttal · Authors · 2025-07-28
>
> We sincerely thank reviewer fL5D for the valuable feedback and appreciation of our contributions! We address the concerns raised as follows.
>
> **The relation of set smoothness to other regularity conditions**
>
> (a) While one of our main contributions is to introduce the concept of set smoothness, we highlight that relating it to the lower-level PL condition is a second key contribution; see Proposition 2. Proposition 2 shows that set smoothness holds for a group of nonconvex-PL BLO problems that are widely studied in the literature. It provides valuable insight into the generality and practical relevance of the set-smoothness condition.
>
> (b) Great suggestion regarding natural examples where set smoothness holds while other assumptions fail. Here is a simple illustrative case: Consider the lower-level function $g(x,y)=h(\sin(x)+y)$ for a function $h:R\to R$ with a nonempty set of minimizers $V=\arg\min_z h(z)$. Then, the solution set function $S(x)=\arg\min_{y}h(x,y)$ satisfies $S(x)=V-\sin(x)$. In this example, $S$ satisfies set smoothness yet no additional regularity assumptions are required on $h$ or $g$. This illustrates that when $g$ has a separate structure in $x$ and $y$, set smoothness may arise naturally. We will incorporate this discussion in the future version of our manuscript in response to your suggestion.
>
> **Preference on Clarke stationarity**
>
> As noted in lines 150-153 of Sec. 2, Goldstein stationarity is a relatively weak condition, whereas Clarke stationarity is significantly stronger. For example,
> there exists a differentiable function $f$ such that a point $x$ with $\|\|\nabla f(x)\|\|\geq1$ still satisfying Goldstein stationarity [31, Proposition 1]. In contrast, Clarke stationarity excludes such points, offering a more meaningful notion of stationarity.
> Consequently, employing Clarke stationarity would lead to much stronger theoretical convergence guarantees for algorithms. Following your suggestion, and given more space in the main body, we will provide a more detailed justification for our preference for Clarke stationarity in a future version of our manuscript.
>
> **Position of set smoothness in broader context**
>
> We agree that discussing the implications of results in Sec. 3 would be highly insightful. As we mentioned in Sec. 5, our results may have potential implication in coupled minmax optimization. Indeed, our Proposition 1 may directly apply to the inner problem of coupled minmax problem (see reference [46]) under set smoothness condition, offering structural insights into such problems.
> For algorithmic implication, the established weak convexity/concavity suggests that the problems fall within the broad class of weakly convex/concave optimization. This connection opens the door to utilizing established methods from that literature, such as subgradient methods, proximal point methods, and zero-order methods.
> We will incorporate these discussion on the implication of our results in the future version of our manuscript.
>
> **Clarification on Algorithm 1 in Sec. 4**
>
> We note that the main contribution of Sec. 4 is establishing the computability of Clarke hyper-stationarity for the concerned BLO, as we stated in lines 293-294 of Sec. 4. To achieve this goal, we do not necessarily need to design a new special algorithm. Instead, we utilize the established structural property in Sec 3 and extend a standard algorithm, i.e., the zero-order method, also used in [7], to both optimistic and pessmistic BLO. While previous works only show that it computes Goldstein stationarity, our analysis provides a stronger convergence guarantee to approximate Clarke stationary points. This is enabled by the weak convexity/concavity we established, as noted in lines 313-315. These results enhance the understanding of the computability of BLO and we leave designation of new efficient algorithms for computing Clarke stationarity for future work, as stated in lines 329-331.
>
> |*Has Definition 2 (set-smoothness) appeared in prior literature, perhaps in the context of set-valued analysis or variational analysis?*
>
> Indeed, we carefully reviewed related concepts for set-valued mappings in the literature on set-valued analysis and variational analysis, including but not limited to the classical reference [28] and a recent book [38]. To the best of our knowledge, the specific concept of set smoothness as defined in Definition 2 has not appeared in prior work. In Footnote 2, we provide a comparison between our proposed notion and existing concepts for set-valued functions. Given more space in the main body, we plan to present this comparison more explicitly in a future version of the paper.
>
> |*Could the proposed method or analysis be extended to stochastic bilevel settings?*
>
> Certainly. We believe it is quite natural to extend our methodology to stochastic/decentralized settings, which are practical for real applications. As the lower-level PL condition has also been adopted in such settings, we expect that similar properties and strong convergence results can be obtained.
>
> If you have any additional concerns regarding our manuscript, we would be happy to provide further clarifications. Finally, we thank the reviewer for pointing out the format issue. We used the provided neurips_2025.sty file in our tex file without modifying the margins or font size. We will address any remaining minor discrepancies in a future version of our manuscript.

---

### Decision · Program_Chairs · 2025-09-17

**Decision:**

Accept (spotlight)

**Comment:**

This paper studies bilevel optimization problems in which the lower level problem does not necessarily admit a unique solution. To circumvent this obstacle, they introduce a notion of set-smoothness for the set of solutions to the lower level problem, which can be used to ensure that the composite function being optimized in the bilevel problem is weakly convex/weakly concave, even if the lower level problem is not convex. With this notion of set-smoothness, they are able to derive zeroth-order algorithms for solving the bilevel optimization problem when the set-smoothness condition holds. They are eventually able to prove convergence to a Clarke stationary point for the bilevel optimization problem with their new assumption.

The paper was well-received by the reviewers, who noted it was well-written, rigorous, and original. The paper could be improved by better explaining the relationship between the set smoothness condition proposed and those that already existed in the literature (e.g., the Aubin property, which is mentioned in the paper but largely skipped over). During the rebuttal period, all reviewers agreed that the paper should be accepted and that any concerns they had had were addressed by the authors.

In the post-rebuttal discussion, I asked reviewers if they felt the paper did a good enough job positioning itself with respect to prior work, specifically referring to the Aubin property as an example. All reviewers found no problem with the submission in this regard so I wholeheartedly recommend acceptance.